



# Topographic disequilibrium, landscape dynamics and active tectonics: an example from the Bhutan Himalayas.

Martine Simoes [1], Timothée Sassolas-Serrayet [2], Rodolphe Cattin [2], Romain Le Roux-Mallouf [2,3], Matthieu Ferry [2], Dowchu Drukpa [4]

[1] Université de Paris, Institut de physique du globe de Paris, CNRS, F-75005 Paris, France.
[2] Géosciences Montpellier, Université de Montpellier, CNRS, Université des Antilles, Montpellier, France.
[3] now at Géolithe, 38920, Crolles, France.
[4] Department of Geology and Mines, Thimphu, Bhutan.

*Correspondence to*: Martine Simoes (simoes@ipgp.fr)

**Abstract.** The quantification of active tectonics from geomorphological and morphometric approaches most often implies that erosion and tectonics have reached a certain balance. Such equilibrium conditions may however be seldom found in nature, as questioned and documented by recent theoretical studies, in particular because drainage basins may be quite dynamic even though tectonic and climatic conditions remain constant. Here, we document this drainage dynamics from the particular case example of the Bhutan Himalayas. Evidence for out-of-equilibrium morphologies have for long been noticed in Bhutan, from

major (> 1 km high) river knickpoints and from the existence of high-altitude low-relief regions within the mountain hinterland. These peculiar morphologies were generally interpreted as representing a recent change in climatic and/or tectonic conditions. To further characterize these morphologies and their dynamics, and from there discuss their origin and meaning, we perform field observations and a detailed quantitative morphometric analysis using χ plots and Gilbert metrics of drainages over various spatial scales, from major Himalayan rivers to local streams draining the low-relief regions. We first find that the river network

is highly dynamic and unstable. Our results emphasize that the morphology of Bhutan does not result from a general wave of incision propagating upstream, as expected from most previous interpretations. Also, the specific spatial organization in which all major knickpoints and low-relief regions are located along a longitudinal band in the Bhutan hinterland, whatever their spatial scale and the dimensions of the associated drainage basins, calls for a common local supporting mechanism most probably related to active tectonic uplift. From there, we discuss previous interpretations of the observed landscape in Bhutan.

Our results emphasize the need for a precise documentation of landscape dynamics and disequilibrium over various spatial scales as a first-order step in morpho-tectonic studies of active landscapes.

## 1 Introduction

The morphology of the Earth's surface and its evolution in space and time result from the competition and balance between tectonics and surface processes. Indeed, the interplay between these processes, with erosion of uplifted terranes

followed by transport and deposition of the produced sediments in subsiding basins, leads to a continuous dynamic



redistribution of masses, eventually modulated by climate (e.g. Allen, 2008). In erosive systems, such as in uplifting and growing mountain ranges, such physical competition between uplift and erosion theoretically leads to steady-state after a characteristic time that mostly depends on the erosional efficiency (e.g. Bonnet and Crave, 2003;Simoes et al., 2010;Whipple and Meade, 2004, 2006;Whipple and Tucker, 1999;Willett et al., 2001). This concept of steady-state and equilibrium between

tectonics and erosion provides an effective conceptual framework, commonly used as a first-order assertion for investigating and quantifying active tectonics from geomorphology (e.g. Lavé and Avouac, 2001). However, topographic equilibrium may eventually only be reached at a regional scale (Willett and Brandon, 2002), and is expected not to be achieved at the more local scale of the drainage network even though tectonic and climatic boundary conditions remain constant, as illustrated in numerical (e.g. Sassolas-Serrayet et al., 2019) or analog (e.g. Hasbargen and Paola, 2000) models. The dynamics of the river

network is expected to be even more pronounced in natural landscapes for which boundary and forcing conditions vary over time and space. In fact, the formalism used in earlier theoretical studies to model erosion is oversimplified as it does not capture the complex 3D dynamic response of drainage basins, in particular through the constant mobility of drainage divides (e.g. Goren et al., 2014;Willett et al., 2014). An example of such landscape dynamics during mountain-building lies in the progressive capture of longitudinal drainages by steeper transverse rivers, as exemplified in the field (e.g. Babault et al., 2012)

or in analog models (Viaplana-Muzas et al., 2015), or in the dynamic re-organization of the river network as a response to tectonic stresses (e.g. Castelltort et al., 2012;Guerit et al., 2018). River captures were also evidenced in old orogens where topographic equilibrium is hypothesized (Prince et al., 2011). This mobility of drainage divides leads to a lengthening of the time needed to reach theoretical steady-state and generates a potential perpetual transience of landscapes (e.g. Hasbargen and Paola, 2000;Whipple et al., 2017b;Yang et al., 2015). It is expected to generate variations in the sedimentation rates at river

outlets by modifying the sediment routing system (e.g. Viaplana-Muzas et al., 2019), and to elucidate observed large dispersions in denudation rates, even in regions that are believed to be in quasi-topographic steady state (Beeson et al., 2017;Sassolas-Serrayet et al., 2019;Willett et al., 2014). Interestingly, this dispersion in measured denudation rates increases most often for smaller drainage basins, further emphasizing that the concept of steady-state is highly scale-dependent as local variations along drainage divides average out at the scale of large river basins (Matmon et al., 2003;Sassolas-Serrayet et al.,

55    2019).

Here, we propose to further investigate these questions, mostly derived from theoretical studies or in few field cases, by considering the emblematic natural example of the Himalayan range in Bhutan (Figure 1). Because of their high rates of continental deformation and erosion, the Himalayas have for long been an ideal natural case for exploring the interactions between tectonics and surface processes, and the associated landscape response (e.g. Adlakha et al., 2013;Beaumont et al.,

2001;Burbank et al., 2003;Clift et al., 2010;Godard et al., 2014;Godard et al., 2004;Grujic et al., 2006;Hodges et al., 2004;Lavé and Avouac, 2001;Thiede et al., 2005;Thiede et al., 2004). In particular, the Himalayas of Central Nepal have been considered as the archetype of an equilibrated topography on average, mostly from their overall concave topography and hypsometry (Duncan et al., 2003), or from the observed first-order consistency between denudation (Godard et al., 2014), incision (Lavé and Avouac, 2001) and exhumation (e.g. Bollinger et al., 2006) rates over various spatial and temporal scales. Such is not the



case further east, in Bhutan, where evidence for out-of-equilibrium morphologies have for long been noticed, from the overall convex topographic profile (Figures 2a-b), high altitude low-relief landscapes (Figure 1c) and major knickpoints (Figure 2c) within the mountain hinterland (Adams et al., 2015;Adams et al., 2016;Baillie and Norbu, 2004;Duncan et al., 2003;Grujic et al., 2006). These morphological features have been interpreted in a variety of ways, mostly as reflecting either climatic (Grujic et al., 2006) or tectonic (Adams et al., 2016;Baillie and Norbu, 2004;Coutand et al., 2014) changes, eventually related to the

uplift of the Shillong Plateau in the foreland further south, or, in other words, as representing relics of former climatic or tectonic conditions and the landscape transience towards a new equilibrium state. However, recent findings in other contexts question these interpretations: low-relief landscapes were shown to form possibly dynamically in-situ as a response to an increase in local base level (e.g. Babault et al., 2007) or to a persistent drainage re-organization, even in the case of constant tectonics and climate (Yang et al., 2015). Finally, despite these observations in Bhutan and the questions they raise, denudation

rates have been considered as reflecting uplift rates and used to derive the geometry of the underlying main active fault (Le Roux-Mallouf et al., 2015) or the timing of interpreted recent tectonic changes (Adams et al., 2016). Whether or not these denudation rates are indeed representative of actual uplift rates needs to be probed. Because smaller drainage basins are the most inclined to have denudation rates deviating from uplift rates in the case of disequilibrium (Sassolas-Serrayet et al., 2019), the answer to this question is intuitively dependent on the size and location of the sampled drainage basins within the overall

drainage system.

**Figure 1** *(next page)***: Topography, relief and geology of Bhutan.**

a) Geological map of Bhutan, after Greenwood et al. (2016). Main tectono-stratigraphic units are from south to north: the Gangetic Plain (GP), the Siwaliks Hills (SH), the Lesser Himalayan Sequence (LHS) including the Paro Metasediments (PM) in Western Bhutan, the Greater

Himalayan Sequence (GHS) including some leucogranites, and the Tethyan Sedimentary Series (TSS). These units are separated by major tectonic contacts, which are from north to south: the Main Frontal Thrust (MFT), the Main Boundary Thrust (MBT), the Main Central Thrust (MCT), and the South Tibetan Detachment (STD). Locally, some other contacts have been described such as the Kakhtang Thrust (KT) in North-Eastern Bhutan. To the north-west of Bhutan, north of the High Range, the Yadong Rift (YR) is part of the South-Tibetan grabens. The extension of the geological map is reported over the shaded topographic map of Figure 1b.

b) Topographic map of Bhutan, from ALOS World 3D – 30m (AW3D30) DEM data. Main drainage basins are delineated by black lines and associated main rivers are color-coded. Thick colored lines correspond to the main trunk rivers, while thinner lines indicate main tributaries. Major knickpoints are also reported along trunk and tributary rivers. Dashed rectangles locate the swath profiles of Figures 2a-b. Orange and green arrows on the side of these rectangles locate physiographic transitions T2 and T3 as placed in Figures 2a-b. Inset: Location of topographic map with respect to regional political borders.

c) Map of local relief, as calculated from the topography shown in Figure 1b, with a moving window of 500 m. Major high-altitude low relief areas are manually delineated by dashed red lines. These are from west to east: the Phobijka (PS), the Bumthang (BS), and the Yarab (YS) surfaces. In Western Bhutan, another region of lower relief is also found in the Bhutan hinterland along the Wang and Amo Chhu rivers, even though less well defined than the others: the Wang surface (WS). From west to east, dashed rectangles locate the extension of Figures S3, 10, 7 and S5 (Figures S3 and S5 in supplementary materials).








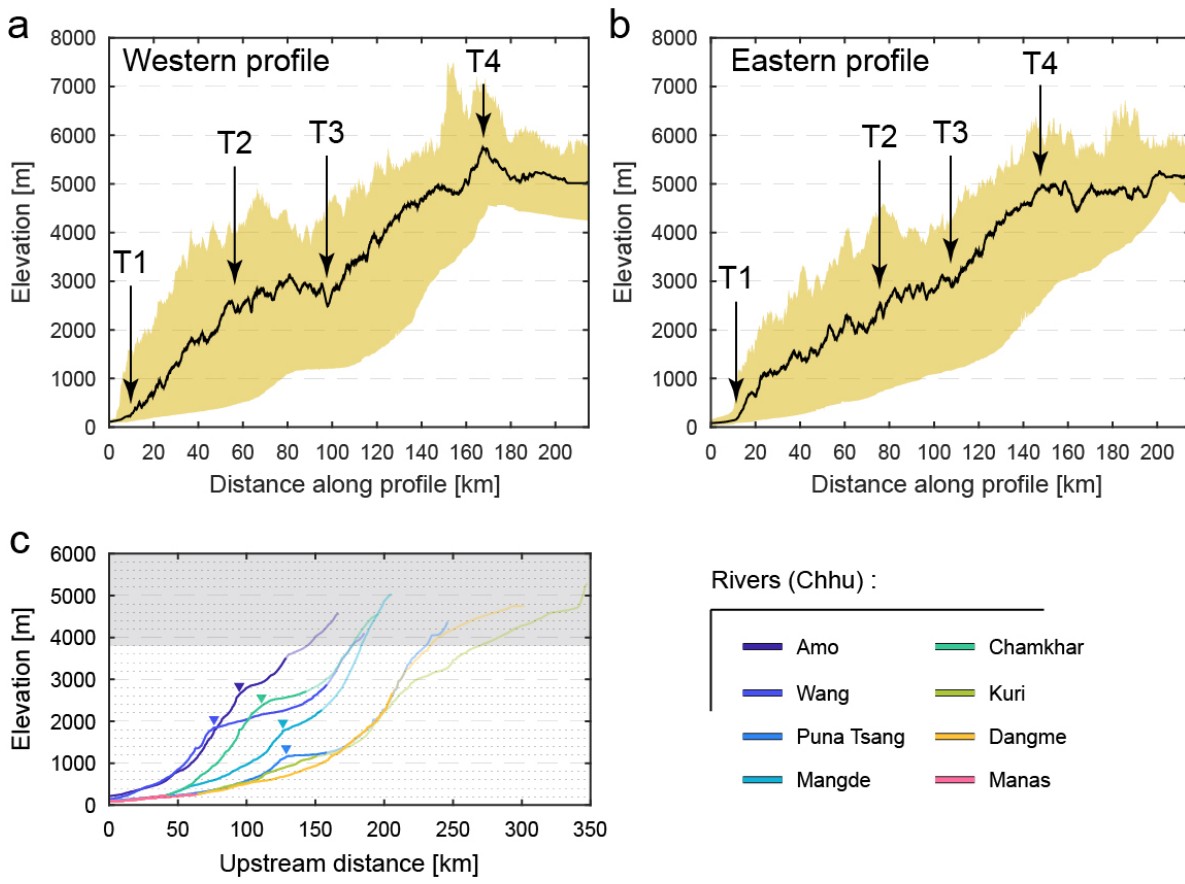

**Figure 2: Topographic and longitudinal river profiles.**

a-b) Swath topographic profiles across the western (a) and eastern (b) Bhutan Himalayas over a 100 km wide region, from the Gangetic Plain up to the Southern Tibetan Plateau. Location of the swaths is reported in Figure 1b. The colored area encompasses the whole range of altitude values, while the black line draws the median altitude across the profiles. Major physiographic transitions, labeled T1 to T4, are also reported.

c) Longitudinal profiles of major and large rivers in Bhutan, illustrating the variety of river profiles. Rivers are located on the maps of Figure 1 and color-coded. Major knickpoints are also pointed by a triangular symbol. Regions with altitudes over 3800 m (grey area) are not to be compared directly to the downstream sections, as these may have a glacial imprint. Portions of the rivers north of physiographic transition T3 are reported by transparent segments.

To get further insights into the landscape dynamics in the Bhutan Himalayas, and in particular further assess the spatial scale dependence of this dynamics, we hereafter conduct a detailed morphometric analysis of this particular field



example. We use χ plots (Perron and Royden, 2013) and Gilbert metrics (Whipple et al., 2017b) for drainages of various dimensions, from major Himalayan rivers to more local streams draining high-altitude low-relief regions. From there, we find

clear evidence for out-of-equilibrium morphologies at various scales, with ubiquitous migrating drainage divides. The geomorphological characteristics of Bhutan do not reveal however a general wave of incision migrating upstream the drainage network. Rather, the co-location and spatial organization of all geomorphic features along a longitudinal swath in the hinterland, whatever the considered spatial scale and the dimensions of the investigated drainage basins, calls for a common local origin and support of the observed landscape dynamics, most probably related to active tectonics. From there we discuss

previous proposed interpretations and the implications of our findings.

## 2 Geological and geomorphological background

### 2.1 Geological setting

The Himalayan arc is characterized by a relative first-order structural and tectono-stratigraphic continuity over c.a.

2500 km (e.g. Gansser, 1983, 1964;McQuarrie et al., 2008). From south to north, and therefore from the Gangetic Plain in the foreland to Southern Tibet, major tectonic contacts are: the Main Frontal Trust (MFT), the Main Boundary Thrust (MBT), the Main Central Thrust (MCT) and the South Tibetan Detachment (STD). The un-metamorphosed synorogenic detrital series of the Siwaliks Group (e.g. Coutand et al., 2016;Gautam and Rösler, 1999) are thrusted over the Gangetic Plain along the MFT, which stands as the southern deformation front of the orogen. The Lesser Himalayan series are thrusted over the Siwaliks

molasses along the MBT. These series are formed of metasediments, initially deposited on the Indian margin during the Proterozoic and Paleozoic (e.g. (Long et al., 2011b;Upreti, 1999) and references therein), and subsequently deformed, metamorphosed and exhumed since the Mid-Late Miocene, mostly in the form of duplexes in the mountain hinterland (e.g. Bollinger et al., 2004;Bollinger et al., 2006;DeCelles et al., 2001;Huygue et al., 2001;Long et al., 2012;Robinson et al., 2001;Schelling and Arita, 1991). Further north, the Greater Himalayan Sequence is thrusted over the Lesser Himalaya along

the MCT. This sequence is constituted of granulites and ortho- and para-gneisses of the former Indian margin, with widespread leucogranites (e.g. Guillot and Le Fort, 1995;Le Fort et al., 1987). The MCT stands as a thick top-to-the-south shear zone, active by Early to Mid-Miocene (e.g. (Tobgay et al., 2012) and references therein), and subsequently deformed by the duplexes of the Lesser Himalayan series. Finally, the Tethyan Sedimentary Sequence is formed of the sedimentary cover of the northern Indian margin (e.g. Liu and Einsele, 1994); it is separated from the high-grade Greater Himalayan rocks by the top-to-the-

north normal STD. Except for the un-metamorphosed Siwaliks, the erodability of the series present throughout the mountain range is found to be comparable to the first-order (Lavé and Avouac, 2001).

The Bhutan Himalayas obey this general tectonic and stratigraphic organization observed throughout the Himalayan arc (Figure 1a), even though presenting some peculiarities and specificities (Gansser, 1983;Greenwood et al., 2016;Long et al., 2011a). In particular, the Greater Himalayan and the Tethyan Sedimentary Sequences are here much more exposed



throughout the mountain belt when compared to the Himalayas of Central Nepal where these sequences are only preserved in the form of individual klippen (e.g. Duncan et al., 2003). In turn, Lesser Himalayan series mostly crop out along the main topographic mountain front, or within structural windows, such as the Paro window in western Bhutan (Tobgay et al., 2012) or the larger Lesser Himalayan window along the Kuri Chhu in eastern Bhutan (Long et al., 2011b) (Figure 1a). Because of the limited flexural subsidence in the broken foreland of Bhutan (Hammer et al., 2013;Verma and Mukhopadhyay, 1977), most

probably related to the presence of the Shillong Plateau basement high further south (e.g. Biswas et al., 2007;Clark and Bilham, 2008;Yin et al., 2010), the Siwaliks series is here much less developed than elsewhere along the Himalayan arc (Hirschmiller et al., 2014), and the MFT is even absent in western Bhutan (Greenwood et al., 2016;Long et al., 2011a) (Figure 1a).

All major thrust faults are interpreted to branch off the Main Himalayan Thrust (MHT) at depth, which stands as the main crustal-scale detachment at the base of the Himalayan orogenic wedge (e.g. DeCelles et al., 2001;Schelling and Arita,

1991). To the first-order the geometry of the MHT is quite simple, with the frontal emerging ramp of the MFT, rooting at depth into a large shallowly north-dipping structural flat, and pursuing northward down to a mid-crustal ramp beneath the high mountain range. This first-order general structure has been proposed throughout the Himalayas, in particular in Nepal (e.g. Bollinger et al., 2004;Lavé and Avouac, 2001;Lyon-Caen and Molnar, 1983, 1985;Schelling and Arita, 1991) but also more recently suggested in Bhutan, as imaged from geophysics (Diehl et al., 2017;Singer et al., 2017), or evidenced from the

modeling of interseismic strain (Marechal et al., 2016), thermochronological data (Coutand et al., 2014) or denudation rates (Le Roux-Mallouf et al., 2015). In the details, the flat structure connecting both the frontal and the mid-crustal ramps could be locally more complex with secondary ramps (e.g. Adams et al., 2016;DeCelles et al., 2001;Hubbard et al., 2016;McQuarrie et al., 2008).

The Himalayan range is actively absorbing shortening, with shortening rates increasing eastwards from c.a. 13 mm/yr

to c.a. 21 mm/yr along the arc (Stevens and Avouac, 2015). Active tectonics are manifested by numerous historical earthquakes all along the Himalayan arc in addition to paleoseismological evidence of major earthquakes rupturing the MHT up to the surface (e.g. (Bilham, 2019) and references therein). Even though much less investigated, the Bhutan Himalaya is also seismically active (Drukpa et al., 2006). It has been struck by a major historical earthquake in 1714 (e.g. Hetenyi et al., 2016), in addition to other past events recently revealed in geomorphological and paleoseismological studies along the frontal MFT-

MBT thrust system (Berthet et al., 2014;Le Roux-Mallouf et al., 2020;Le Roux-Mallouf et al., 2016).

## 2.2 Geomorphology of Bhutan: general characteristics

The topography of the Bhutan Himalayas is characterized by a convex profile, with four major physiographic transitions, hereafter referred as T1 to T4 from south to north (modified after Adams et al. (2013), Baillie and Norbu (2004),

Duncan et al. (2003)) (Figures 2a-b). The first one (T1) corresponds to the mountain topographic front, from where the mountain range rises above the Indo-Gangetic Plain. Northward from the mountain front, topography overall increases continuously up to c.a. 3000 m on average for the first c.a. 70-80 km (T2), then remains constant on average and high, increases



again after c.a. 100-125 km (T3) and reaches average values of c.a. 5000 m by c.a. 175 km (T4) and northwards (Figures 2a-b). Except for T2, these physiographic transitions are also existent in Central Nepal, where the abrupt topographic rise between

T3 and T4 has been interpreted as related to the uplift of the High Himalayan range above the mid-crustal ramp of the MHT (e.g. Bollinger et al., 2006;Cattin and Avouac, 2000;Lavé and Avouac, 2001;Lyon-Caen and Molnar, 1983, 1985). This interpretation is also expected to hold in Bhutan, as this elevation ascent coincides spatially with the location of the recently evidenced mid-crustal ramp (Coutand et al., 2014;Diehl et al., 2017;Marechal et al., 2016;Singer et al., 2017) - or eventually slightly deported south of this ramp (Le Roux-Mallouf et al., 2015).

190        The abrupt topographic rise from T1 to T2 and the high-altitude region between T2 and T3 are mostly specific to the landscape of the Bhutan Himalaya along the Himalayan arc (Adams et al., 2013;Duncan et al., 2003). The region between T2 and T3 is characterized by a locally relatively lower relief (Figures 1c and 3a-d), in contrast to what is observed further south from T1 to T2 (Figures 3e-f) but also further north from T3 to T4 where high elevation crests are flanked by deep gorges (Adams et al., 2015;Adams et al., 2016;Baillie and Norbu, 2004;Duncan et al., 2003;Grujic et al., 2006) (Figure 1). More

precisely, these lower relief regions may correspond either to high-elevation regions around low-relief alluvial valleys drained by major rivers (ex: the Wang and Bumthang regions traversed by the Wang and Chamkhar Chhu rivers, respectively) (Figures 1c and 3a-b), or to smaller elevated low-relief landscape patches connected to the main drainage network through secondary streams (ex: Phobijkha or Yarab surfaces) (Figures 1c and 3c-d). These valleys or surfaces are characterized by stable soils with saprolite horizons and locally contain bogs (Adams et al., 2016;Baillie et al., 2004). It should be noticed that high-

elevation low-relief alluvial valleys found along some of the Himalayan rivers (Figures 3a-b) act as local base levels for upstream erosion.

Figure 3 *(next page)*: **Field pictures representative of the various landscapes in the hinterland of the Bhutan Himalaya**.

Topographic map in the center locates field pictures relative to low-relief regions (contoured with red dashed line) and to major knickpoints along rivers (rivers color-coded as in Figures 1-2).

a) Alluvial plain along the PunaTsang Chhu, upstream of its major knickpoint. Picture taken nearby the Dzong of Punakha.

b) Alluvial plain along the Chamkhar Chhu (Bumthang surface), upstream of its major knickpoint. Picture taken nearby Jakar.

c) High-altitude low-relief area of Ura (south-easternmost portion of the Bumthang surface).

d) High-altitude low-relief area of Kotakha (westernmost portion of the Phobijka surface).

e) Deep gorge south of the major knickpoint along the Chamkhar Chhu.

f) Canyon along the Kuri Chhu.






Major rivers in Bhutan have their sources along the southern flank of the high Himalayan peaks, except for the trans-Himalayan Kuri and Dangme Chhu rivers to the east, and to some extent the Amo Chhu to the west, whose headwaters locate

within Southern Tibet. These rivers are characterized by a variety of profiles (Baillie and Norbu, 2004), but most of them have significant knickpoints, in some cases of the order of - or higher than - c.a. 1 km (Figure 2c). These knickpoints occur at the head of gorges flowing through low-relief landscapes or just downstream of them (Adams et al., 2016;Baillie and Norbu, 2004)



(Figures 1 and 3e). They separate rivers in two main segments (Figure 2c): river profiles are steep - and most probably over-steepened with respect to tectonic uplift (Adams et al., 2016) - downstream of these knickpoints, while upstream river portions

have alluvial fills (Figures 3 a-b) and locate in the low-relief portions of the mountain range, north of T2.

Glacial valleys and cirques are mostly restricted to altitudes higher than c.a. 4200 m (Iwata et al., 2002), either nearby T2 or mostly north of T3, and may have advanced as much down as to altitudes of c.a. 3800 m during the Holocene and Pleistocene (Iwata et al., 2002;Meyer et al., 2009). There is no evidence for glacial imprint on the high-altitude low relief landscapes investigated here, except locally along some of the southern edges of low relief areas where altitudes reach the c.a.

4000 m (Adams et al., 2016).

## 2.3 Geomorphology of Bhutan: previous interpretations

By comparing the morphology and geology and these two regions of the Himalaya, Duncan et al. (2003) proposed that first-order topographic and morphologic differences between Central Nepal and Bhutan could be related to an along-strike

tectonic segmentation of the Himalayan range and associated variable balance between uplift and erosion, in which the timing of deformation and uplift would be younger in Bhutan, leading to a less mature landscape. This interpretation would be supported by the extensive exposure of Great Himalayan and Tethyan sequences in Bhutan (e.g. Gansser, 1983;Greenwood et al., 2016;Long et al., 2011a) (Figure 1a), which contrasts with the klippen of Central Nepal (e.g. Bollinger et al., 2004;DeCelles et al., 2001;Gansser, 1964;Schelling and Arita, 1991). Lateral variations in the timing of major fault systems along the

Himalayan arc cannot be excluded, such as for the Main Central Thrust by the Early to Mid-Miocene (e.g. (Tobgay et al., 2012) and references therein) or for the exhumation of the Lesser Himalayan sequences since the Late Miocene (e.g. (Bollinger et al., 2006;Huygue et al., 2001;Long et al., 2012;Robinson et al., 2001) and references therein). However, such interpretation would first require that the large-scale landscape response time be much greater than what could be justifiable given the regional high erosion rates (e.g. Lavé and Avouac, 2001). Also, in this case, erosion rates in Bhutan would be expected to have

increased in recent geological times, as a transient response to balance uplift. However, thermochronological data rather suggest a general decrease in exhumation rates over the last 4-6 Myr (Adams et al., 2015;Coutand et al., 2014;Grujic et al., 2006).

This general decrease in exhumation rates since Late Miocene - Early Pliocene has been invoked by Grujic et al. (2006) to be related to the rainshadow generated by the coeval topographic rise of the Shillong Plateau further south in the foreland of

Bhutan (e.g. (Biswas et al., 2007;Clark and Bilham, 2008;Govin et al., 2018;Rosenkranz et al., 2018) and references therein). Such rainshadow would have led to lower erosion rates, and from there to locally increase relief and elevation by modifying the balance between uplift and erosion. Following this idea, high altitude low-relief areas in Bhutan would be relict landscapes of prior wetter conditions. However, as noticed by Baillie and Norbu (2004), the rainshadow by the Shillong Plateau is rather relative as first-order climatic conditions are everywhere wet and tropical along Bhutan, as further eastward or westward

(Bookhagen and Burbank, 2006). Some variations exist along the foothills (from physiographic transitions T1 to T2), but climatic conditions are rather similar within the range interior (north of T2). Also, low-relief surfaces are mostly found within





the western half of Bhutan (Figure 1c), where the Shillong rainshadow is supposedly lowest or inexistent according to (Grujic et al., 2006).

In addition to these general observations on the overall topography or the presence of low-relief areas in the range interior, Baillie and Norbu (2004) noticed the variety of river profiles (Figure 2c), despite overall similar first-order climatic and lithologic conditions within Bhutan, and from there suggested that these morphologies be mostly related to lateral differential tectonics, accommodated by north-south faulting. However, such laterally variable uplift should produce a significant along-strike variability in exhumation rates, a pattern not seen in thermochronological data (Adams et al., 2015;Coutand et al., 2014;Grujic et al., 2006). Indeed, most significant variations in exhumation rates are across-strike and not

along-strike, and should relate to cross-sectional changes in the geometry of the underlying Main Himalayan Thrust (Coutand et al., 2014;Le Roux-Mallouf et al., 2015).

More recently, Adams et al. (2015) found that the decrease in exhumation rates since 4-6 Ma retrieved from thermochronology is observed everywhere in Bhutan, within or outside a supposed rainshadow, and proposed that it reflects a southward transfer of deformation towards the Shillong Plateau. Morphological characteristics in Bhutan are here interpreted

as a recent rejuvenation of the landscape related to uplift in the mountain hinterland that would not have yet produced sufficient exhumation to be recorded in cooling ages. Based on the observation that large rivers are over-steepened between physiographic transitions T1 and T2 (Figure 2c) and that most of these rivers develop alluvial plains upstream of T2 (Figures 3a-b), these authors subsequently proposed that major convex knickpoints and upstream low-relief landscapes around alluvial valleys are the geomorphic response to the uplift related to a local blind duplex in the Bhutan hinterland (Adams et al., 2016).

This interpretation also relies on the similarity between these observations and the results of a numerical landscape evolution model (using the CHILD code (Tucker et al., 2001)) in which rivers adjust to higher uplift rates in the hinterland of a mountain range by aggrading upstream of migrating convex knickpoints. Major knickpoints are interpreted by Adams et al. (2016) as migrating upstream and upwards, and while so as removing packages of fill deposits in the upstream alluvial valleys. Failure to follow the rising base level set by these migrating knickpoints would lead to hanging and internally drained valleys, by

analogy to observations in other settings or to simulations of landscape response to uplift (e.g. Burbank et al., 1996;Crosby et al., 2007;Gasparini et al., 2007;Humphrey and Konrad, 2000). Within this framework, low-relief hanging fill valleys can be interpreted as relict landscapes formed locally in the hinterland, and from there can be used to derive the amount of uplift and the timing of the tectonic perturbation (Adams et al., 2016). Using denudation rates measured from in situ produced cosmogenic isotopes in river sands, within and outside the low-relief areas, Adams et al. (2016) suggest that this rejuvenation

is c.a. 0.8-1 Ma old. Even though the idea of localized uplift over a blind ramp is seducing, some details in their interpretation need further investigations. Indeed, given the variety of river profiles in Bhutan (Figure 2c), the choice of the river impacts the amount of uplift derived by comparing the actual profile with a theoretical one. Also, using differential denudation rates to determine the timing of uplift implies that these are representative of uplift rates, an assertion that may not be valid in a setting where the landscape is inevitably out-of-equilibrium (e.g. Sassolas-Serrayet et al., 2019).





All previous interpretations have therefore attributed the specificities of the morphology of the Bhutan Himalayas (by contrast to that of Central Nepal), to either changes in climatic or tectonic boundary conditions related in most cases in a way or the other to the uplift of the Shillong Plateau in the foreland of Bhutan. All interpretations rely on the idea that high-altitude low-relief regions are relict landscapes of previous conditions, and that these features as well as major knickpoints reflect a general transient wave of incision migrating upstream in the drainage system. These various assertions on the dynamics of the

river network still need however to be clearly documented and verified.

## 3 Data and Methods

        Field observations within the hinterland of Bhutan are complemented by a morphometric analysis of the landscape dynamics, further detailed hereafter. Field work was conducted during two two-weeks long campaigns in January-February

2015 and in February 2017.

### 3.1 Topography: data and analytical tools

        Our study relies on topographic data obtained from the ALOS World 3D – 30m (AW3D30) Digital Elevation Model (DEM) provided by the Japanese Aerospace Exploration Agency. It has been shown that this DEM has the highest precision

when compared to other DEMs with equivalent resolution so that it is well suited to the analysis of river profiles in high relief regions (Schwanghart and Scherler, 2017). For our analysis, we use the Matlab-based function library TopoToolbox, which provides essential tools for large scale geomorphological approaches, including drainage network and watershed extraction, or computation of usual river or topography metrics (Schwanghart and Scherler, 2014). For additional information, in particular on the various algorithms of the library used hereafter, the reader is referred to the TopoToolbox documentation

(https://topotoolbox.wordpress.com) and to the references provided therein.

        Topographic relief is determined as the difference between maximum and minimum altitudes in a sliding window of 500 m radius so as to get a fine resolution on this metrics. Even though we extract morphometric parameters along the whole drainage network, we specifically focus on the region south of the High Himalayan peaks, i.e. south of physiographic transition T3, where the fluvial network is supposedly the least affected by a potential glacial imprint (altitudes < 3 800 m) and where

morphological features characteristic of disequilibrium (major knickpoints, elevated low-relief regions) have been described.

### 3.2 Drainage network analysis

### 3.2.1 Drainage network extraction and pre-processing

        To extract the hydrographic network, we use the Single Flow Direction algorithm (SFD) implemented in

TopoToolbox, from which an accumulation map is computed. This accumulation map provides the drainage area upstream





each pixel of the DEM, and from there allows to choose the network extension by defining a threshold source area. We extract the river network with a minimal drainage area of 50 km². This systematic extraction avoids the manual *ad-hoc* selection of specific rivers, which could otherwise bias the analysis.

325       We extract the main river systems of the Bhutan Himalaya upstream of the topographic front (T1). This supposes that the Gangetic plain is taken as the base level of the extracted upstream drainage network. In Eastern Bhutan, major river confluences occur just upstream the deformation front: indeed, the Manas Chhu river forms downstream of the confluence of the four Mangde, Chamkhar, Kuri and Dangme Chhu transhimalayan rivers from west to east, respectively (Figure 1). In this specific case, we individualize these river systems upstream of their confluences to improve the clarity of the figures and to facilitate their comparison.

330       Once the river network is extracted, we erase small scale topographic noise associated with DEM artifacts along river paths using the Constrained Regularized Smoothing (CRS) algorithm (see (Schwanghart and Scherler, 2017) for further details). Smoothing parameters are carefully selected by cross-checking so as to remove erratic noise while keeping real topographic features such as knickpoints that may be present along river profiles. The use of smoothed river profiles allows for an easier extraction of knickpoints. It also allows for a more accurate computation of stream metrics such as χ.


### 3.2.2 Extraction of knickpoints

      Knickpoints are automatically detected using the Knickpointfinder algorithm implemented in TopoToolbox (Schwanghart and Scherler, 2014). As for the smoothing of river profiles, detection parameters are refined by cross-checking. The final knickpoints list is then manually corrected. We do not consider knickpoints above altitudes of 3800 m as these may

be mostly due to glacial or post-glacial morphologies. We also manually remove doubtful minor knickpoints most probably associated with imperfect local smoothing. This method does obviously not give an exhaustive list of knickpoints along Bhutanese rivers, but it ensures the retrieval of large knickpoints and of their spatial organization as needed for our argumentation.

**3.2.3 Computation of transformed coordinates (χ)**

      The computation of χ transformed river coordinates has proved useful for the understanding of landscape and drainage network dynamics (e.g. Perron and Royden, 2013;Whipple et al., 2017b;Willett et al., 2014;Yang et al., 2015). In this coordinate system, the elevation $z$ at a given position along the river bed $x$ depends of the integral quantity χ following equation (1):

$$z(x) = z(x_b) + \left(\frac{U}{KA_0{}^m}\right)^{\frac{1}{n}} \chi \tag{1}$$

with

$$\chi = \int_{x_b}^{x} \left(\frac{A_0}{A(x)}\right)^{\frac{m}{n}} dx$$



where $x_b$ locates the channel outlet; $A(x)$ is the drainage area upstream of the position $x$ along the channel; $A_0$ is a reference scaling area; $m$ and $n$ are constant empirical parameters most often expressed as the $m/n$ ratio. This ratio can be related to the

reference concavity index $\theta_{ref}$ that describes how the river gradient evolves along the river profile. Such integral approach is based on the stream power equation of river incision (e.g. Howard, 1994;Whipple and Tucker, 1999). It provides a simple metric for pointing out and for analyzing the nature of transient signals within the river network, by comparing trunk streams and their tributaries in profiles corrected for their variable upstream drainage areas  - and therefore their variable spatial scales - (Perron and Royden, 2013) (Figure 4).


**Figure 4** *(next page)*: **Schematic river profiles and metrics expected in the case of equilibrated rivers, of a wave of incision migrating upstream the river network or of stream piracy and divide migration.**

a) Longitudinal (left) and χ (right) profiles of a main trunk river (dark line) and its tributaries (grey lines) in the case that these rivers are in

equilibrium with external forcing conditions that are constant over the whole area. The χ profiles of these various rivers are all colinear, within a certain variability related to natural conditions (Perron and Royden, 2013).

b) Longitudinal (left) and χ (right) profiles of a main trunk river (dark line) and its tributaries (grey lines) in the case of the migration of a wave of incision through the river network. Knickpoints are present along the various streams at variable locations, but their χ profiles are all concordant, within the variability due to natural conditions (Perron and Royden, 2013).

c) χ profiles expected in the case of migrating divides (left) or river captures (right). Aggressor (or pirate) stream is represented in red, and the victim in blue. The dark line represents the profile expected for rivers equilibrated with external forcing. Area gain shifts the χ profiles of aggressor streams to lower χ values, above the equilibrium line. Conversely, area loss shifts the χ profiles of victim streams to higher χ values, below the equilibrium line. Here, χ profiles are discordant (Willett et al., 2014).

d) Assessment of divide migration and its direction from the across-divide contrast in χ (Willett et al., 2014) (left)  or in Gilbert metrics such

as local slope or relief  (Forte and Whipple, 2018;Whipple et al., 2017b) (right).







## Basics of chi transformed profiles

**a**

**Equilibrated river profile**

**b**

**Migrating incision wave**

## Morphometric tools to assess drainage piracy

**c**

On profile

- Victim
- Aggressor
- Equilibrated profile

- Victim
- Aggressor
- Captured reach

**d**

On map

$\chi$ or elevation value

Low | High

Local slope or relief value

Low | High



The χ plot of an equilibrated concave river system obeying the stream power equation, with uniform conditions, is a straight line along which both main trunk stream and tributaries collapse (Figure 4a). Furthermore, by setting $A_0$ to unity, the slope of the χ profile is equal to the channel steepness $K_{sn}$ (Perron and Royden, 2013). This metric is linked to the uplift rate $U$ and the erodibility coefficient $K$, by the equation:

$$K_{sn} = \left(\frac{U}{K}\right)^{\frac{1}{n}} \tag{2}$$

For $U$ and $K$ varying along the profile of the river, steeper (gentler) segments in χ profiles relate either to locally higher (lower) uplift rate or to lower (higher) erodibility. However, any quantification of these parameters from river profiles is not direct and straightforward as they depend on parameter $n$, which is not known a priori (e.g. Mudd et al., 2018). In the following, we assume that there is no significant variation in the erosion processes over the study region. We also do not consider a specific value for $n$ and use $\theta_{ref} = 0.5$ (see supplementary material, in which the effect of the chosen concavity index is tested for $\theta_{ref}$

ranging between 0.3 and 0.7). Such assumptions allow for a relative comparison between the various parts of the drainage network over our study area.

### 3.2.4 Assessment of the drainage network dynamics

We use transformed χ profiles to detect and analyze perturbations within the river network. Knickpoints in long-

distance profiles of main trunk and tributaries that migrate upstream as a response to a common process and wave of incision are expected to overlap and be concordant in transformed coordinates (Perron and Royden, 2013) (Figure 4b). Conversely, migrating knickpoints of differing origins (ex: local discrete river captures) will be discordant both in longitudinal and χ profiles. In the case of migrating knickpoints, whether concordant or discordant in χ coordinates, the river steepness around knickpoints is representative neither of the actual local uplift rate nor of the erodibility, but rather of the upstream migration

of an incision wave. Spatially fixed knickpoints, as a response to local variations in uplift or erodability, are expected to be also discordant in transformed coordinates.

Transformed χ plots are also useful to unravel drainage reorganization related to migrating divides and river captures (Willett et al., 2014). A river capture leads to a characteristic χ profile (Willett et al., 2014) (Figure 4c). Indeed, an abrupt increase in drainage area due to a capture shifts the profile of the pirate stream to lower χ values, i.e. above a theoretical

equilibrium profile. Conversely, area loss shifts the profile of the victim stream to higher χ values, i.e. below a theoretical equilibrium profile (Figure 4c). χ profiles of rivers affected by captures are therefore expected to be clearly discordant. This is also expected for migrating divides without discrete captures (Yang et al., 2015), unless the response of channels to drainage area changes is faster than that of divide migration (Whipple et al., 2017b).

We also use maps of χ along the drainage network to detect potential migrating divides, and to assess the direction of

these migrations. A contrast in the value of χ across a divide results in its migration toward the stream with the highest χ (Willett et al., 2014) (Figure 4d). Recent studies show however that the direction of divide migration interpreted from a χ



analysis is not straightforward and can be biased by the assumed river base level (Whipple et al., 2017b). Spatially variable tectonic or climatic conditions, as well as rock properties, may also maintain χ contrasts across stable divides. To overcome these limitations, we strengthen our analysis of across-divide contrasts by using "Gilbert metrics" associated with contrasts in elevation of the river bed, in mean upstream local slope and in mean upstream local relief at a given drainage reference area (Forte and Whipple, 2018;Whipple et al., 2017b), taken here as 2 km². Migration of the divide is expected to occur towards the stream with relative higher elevation, and conversely with relative lower local slope or relief (Figure 4d).

In addition, we infer divide motion by locating morphological indicators of regressive erosion like beheaded channels (or wind gaps), from Google Earth satellite imagery or field observations. The use of these complementary methods allows us to produce a more careful assessment of divide migration directions, and thus of the actual drainage network dynamics.

## 4 Results

### 4.1 General observations, and definition of the spatial scales of investigation

Altogether, the analysis of longitudinal and χ profiles suggest three distinct sets for the major rivers in Bhutan (Figures 2c and 5). First, the Kuri and Dangme Chhu in Eastern Bhutan are characterized by relatively simple profiles south of the high range (south of physiographic transition T3), without any remarkable major knickpoint. These rivers correspond to the two largest drainage basins, with drainage areas > 9000 km$^2$ (Table 1). Second, rivers like the Amo, Wang and Chamkhar Chhu have major knickpoints (> 1 km high), located approximately nearby T2. These knickpoints separate a steep river segment to the south from a river segment with lower gradient to the north. More specifically, the river portion with lower gradient flows within and across an alluvial plain (Figure 3b) located in a high-altitude low-relief region (Figure 1c). These rivers are those among the major Himalayan rivers of Bhutan with the smallest drainage basins (drainage areas of < 5000 km$^2$) (Figure 1, Table 1 and Figure S8 in supplementary material). Third, in between these two sets, the Puna Tsang and Mangde Chhu have intermediate characteristics and longitudinal χ profiles. Indeed, they show a more modest (< 1 km high) knickpoint nearby the region of T2, and flow through a limited (Puna Tsang Chhu - Figure 3a) or inexistent (Mangde Chhu) alluvial plain upstream of this knickpoint.





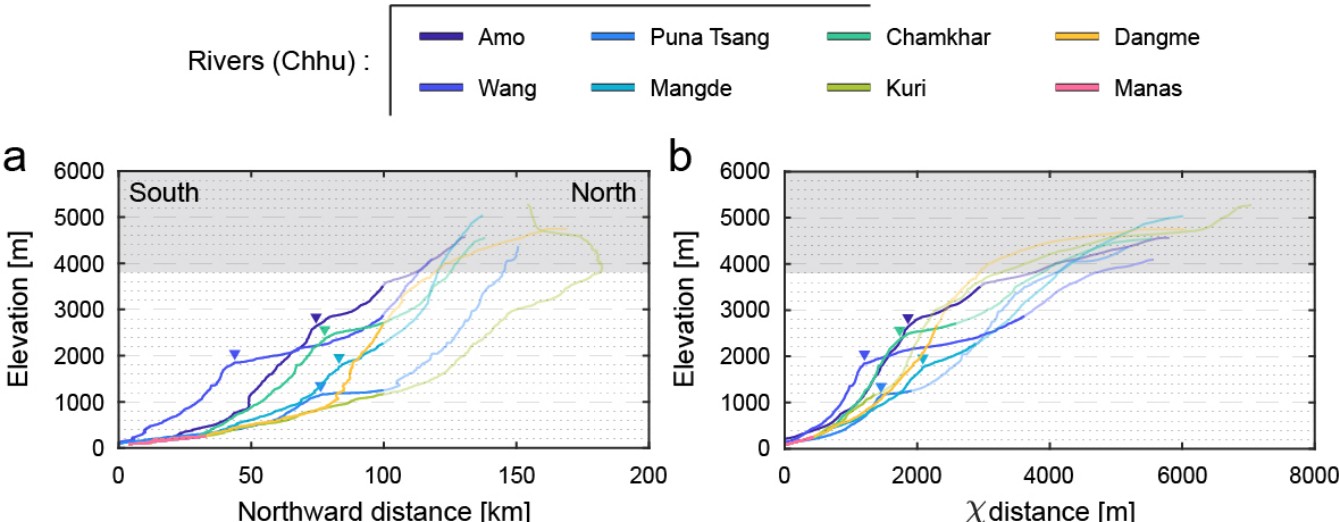

**Figure 5: Projected and transformed profiles of major and large rivers in Bhutan.**

Rivers are located on the maps of Figure 1 and color-coded. Major knickpoints are also pointed by a triangular symbol. Regions with altitudes over 3800 m (grey area) are not to be compared directly to the downstream sections as these may have a glacial imprint. Portions of the rivers north of physiographic transition T3 are reported by transparent segments.

a) Longitudinal profiles projected along a north-south axis, approximately perpendicular to active tectonic structures and to structural directions (Figure 1a). Except for the Wang Chhu, all major knickpoints are located c.a. 75-80 km north of the mountain front.

b) Transformed river profiles, following the formalism of Perron and Royden (2013). The Gangetic Plain is used as the base level for all these rivers. $\chi$ transformed profiles are established with a concavity of 0.5 (see Figure S1 in supplementary material for similar transformed profiles but using other concavity values).

**Table 1: Drainage areas of large Himalayan rivers in Bhutan, taken upstream of their outlet at the mountain front or upstream of**
**a major confluence.**

| River | Drainage area (km²) |
|---|---|
| Amo Chhu | 3752 |
| Wang Chhu | 4600 |
| Puna Tsang Chhu | 9701 |
| Mangde Chhu [a] | 3819 |
| Chamkhar Chhu [a] | 3174 |
| Kuri Chhu [b] | 9652 |
| Dangme Chhu [b] | 10485 |

[a] The drainage area of the Mangde and Chamkhar Chhu is here calculated upstream of their confluence.

[b] The drainage area of the Kuri and Dangme Chhu is here calculated upstream of their confluence.



455        These major rivers define the local base level for the incision of their tributaries (Figure 6). Longitudinal profiles of tributaries exhibit a great variability relative to their trunk stream (Figure 6a). They are better organized when transformed into $\chi$ coordinates (Figure 6b). On one hand, most tributaries appear colinear with their trunk stream in $\chi$ plots - at least colinear in $\chi$ over comparable spatial regions of the mountain range, in particular at the level of their confluence -, indicating that these tributaries clearly follow the local incision pace imposed by the main river. On the other hand, some tributaries clearly plot

above the trunk stream. These tributaries are characterized by a downstream steep channel and an upstream gentle segment. The gentle segments correspond to high-altitude low-relief patches of the landscape (Figures 1 and 3c-d).

       Given the above first-order observations of major rivers and of their tributaries, we define various scales of investigation of the landscape dynamics in Bhutan. First, we analyze major trunk rivers draining the Bhutan Himalayas and discuss their diversity. Next, we focus on those showing major knickpoints and flowing through alluvial plains in high-altitude

low-relief regions of the range. We finally analyze the dynamics of the tributaries of main trunk rivers, in particular nearby perched low-relief surfaces. We recall that we focus hereinafter only on the regions south of physiographic transition T3.

## 4.2 Major Himalayan rivers

       Here, the longitudinal and $\chi$ profiles of the Kuri, Dangme, Puna Tsang and Mangde Chhu are discussed and compared

(Figures 2c and 5). Except for the Mangde Chhu, these rivers correspond to the largest drainage basins of Bhutan (Table 1).

       The two easternmost major rivers (Kuri and Dangme Chhu) show a very similar long river profile, incising deep into the mountain range (Figure 3f) as altitudes of c.a. 2000 m are only reached c.a. 200 km from their outlet at the Gangetic Plain (Figure 2c). This is also the case for the Puna Tsang Chhu, c.a. 100-150 km further west in Western Bhutan, which only departs locally from the previous long profiles by c.a. 130 km from the outlet, at the level of its major - but relatively modest (c.a. 300-

400 m high) - knickpoint. The long profile of the Mangde Chhu is well above that of these other major rivers.

       When transformed into $\chi$ coordinates, the profiles of these four rivers compare well and are colinear to the very first-order within our region of interest south of T3 (Figure 5b). Altogether, these observations suggest that these rivers share and have adjusted to the first order to similar tectonic and/or climatic forcing conditions, even though located throughout Bhutan (Figure 1). It should be emphasized here that the major knickpoints of the Puna Tsang and Mangde Chhu are not concordant

in $\chi$ coordinates.









**Figure 6** *(previous page)*: **Longitudinal (a- top) and transformed χ (b- bottom) profiles of major rivers of Bhutan and of their tributaries** (with drainage area > 50 km2) (location in Figure 1). Major drainage basins (and the corresponding horizontal axes of their profiles) are color-coded as in Figure 1. Trunk streams are reported in bold lines, and tributaries in thinner lines. Lighter transparent colors are used along river profiles for the river portions north of physiographic transition T3. Major knickpoints are pointed out by triangular symbols. Altitudes of 3800 m are considered aside (grey band) as rivers may preserve a glacial imprint in these regions. χ transformed profiles are established with a concavity of 0.5. Other concavities have been tested and are illustrated in Figure S2 (supplementary material). For an easier reading of the figure, horizontal axes (distance or χ) are alternatively reported below or above the graphs, and follow the same color-code as that of the rivers they are associated to.

### 4.3 Large Himalayan rivers draining low-relief alluvial plains

The Chamkhar, Wang and Amo Chhu have either longitudinal or χ profiles that are clearly above those of the formerly discussed major Bhutanese rivers (Figures 2c and 5b). This may not be surprising for longitudinal profiles as these three rivers have more modest drainage areas (Table 1). This pattern remains even in χ coordinates, clearly suggesting that these rivers either face different tectonics and/or climate within the range interior or that they have not yet equilibrated to these forcing conditions. In the case of the Chamkhar Chhu, common first-order forcing conditions are expected to be shared with other nearby major rivers (Mangde Chhu to the west, and Kuri Chhu to the east). Given this, we favor here the idea that its profile is representative of a disequilibrium.

Such χ profiles, well above the regional average defined by the profiles of other major rivers (Figure 5b), are often interpreted as reflecting disequilibrium with a gain of drainage area by stream piracy (e.g. Willett et al., 2014;Yang et al., 2015) (Figure 4c). These profiles would suggest the existence of dramatic river captures in Bhutan, with > 1 km high knickpoints and at the scale of the whole large low-relief (and low-steepness) regions of the Chamkhar and possibly the Wang and Amo Chhu.

The disequilibrium of these rivers is also reflected in the mobility of their drainage divides, as illustrated in Figures 7 and 8 in the case of the Chamkhar Chhu. As intuitively expected, the high-altitude low-relief Bumthang region traversed by the Chamkhar Chhu is being laterally aggressed by the tributaries of the deeply incising Mangde (to the west) and Kuri (to the east) Chhu. As a result, the main drainage divides around this low-relief region are migrating inwards and drainage area is locally shrinking (Figures 7 and 8). The reverse situation is observed further south, downstream of the major knickpoint of the Chamkhar Chhu. Gilbert metrics suggest that drainage divides are here migrating outwards so that drainage area is locally increasing (Figure 7). Similar observations and conclusions are reached for the Wang Chhu (see Figure S3 in supplementary material), even though the situation of the Wang Chhu is slightly more complex.





**Figure 7: Assessing divide mobility around the high-altitude low-relief region of the Chamkhar Chhu** (see Figure 1c for location), as

in theoretical Figure 4d. Map represents local relief, with a moving window of 500 m, and the identified low-relief region is delimited by

the red dashed line. Major knickpoints are reported with inversed triangular symbols and arrows locate the existence of morphological

observations of regressive erosion (wind gaps), together with the direction of the associated divide migration. Similar observations are

reported for the Wang surface (Figure S3 in supplementary material).

a) χ along the river network as a criterium to assess divide mobility (following Willett et al. (2014)). The values of χ are determined for a

concavity of 0.5.

b) Stream elevation as a criterium to assess divide mobility (following Forte and Whipple (2018) and Whipple et al. (2017b)).







**Figure 8** *(previous page)***: Satellite and field observations on the morphological contrasts across and around the Bumthang surface** (location on Figure 1c). Central map reports relief and χ along the river network as in Figure 7a. Satellite observations are taken from the Google Earth database.

a) ©Google Earth view (Image from CNES - Airbus 2020) of the relief and elevation contrasts across the western drainage divide of the Bumthang surface. White arrows locate observed beheaded channels (wind gaps), and indicate the direction of deduced drainage divide
migration.

b) Field picture of the Sengor area (southeastern part of the Bumthang surface), where a filled valley is surrounded by higher relief rims. The filled valley is connected to the main river network by a tributary of the Kuri Chhu. A major knickpoint is present along this tributary, downstream of the pictured valley. Note the contrast in relief with picture d), across the local delimitation of the Bumthang surface.

c) ©Google Earth view (Image from CNES - Airbus 2020 and Landsat - Copernicus) of the relief and elevation contrasts across the eastern
drainage divide of the Bumthang surface. White arrows locate observed beheaded channels, and indicate the direction of deduced drainage divide migration.

d) Field picture illustrating the extremely steep hillslopes along the western Kuri Chhu valley, immediately east of the Bumthang surface. A person is standing on the lower right side of the picture for scale (red circle). Note the contrast in relief with picture b), across the local delimitation of the Bumthang surface.


Altogether, our results indicate that the Chamkhar Chhu, and possibly the Wang Chhu, flowing through high-altitude low-relief alluvial plains in the hinterland (Figures 3b and 8), have overall disequilibrium profiles possibly reflecting the existence of dramatic river captures. Additionally, in the details, we find evidence for an ongoing dynamic rearrangement of
the river network within these large drainage basins, with a specific pattern of drainage area contraction and expansion on either side of the major knickpoint of these rivers. Finally, the profiles and the major knickpoints of the Chamkhar, Amo and Wang Chhu rivers are not colinear in χ coordinates (Figure 5b).

## 4.4 Low-relief regions captured by secondary streams

At a more local spatial scale, modest high-altitude low-relief regions (Figures 3c-d) are connected to the main drainage network through secondary streams and tributaries, as for the Phobijka and Yarab surfaces (Figures 1c).

Starting from the profiles of Figure 6, we further explore the main features of the drainage network within and around the Phobijka surface using longitudinal and χ profiles of the secondary streams, with respect to their main trunk rivers, the Puna Tsang and Mangde Chhu (Figure 9). We here follow the approach proposed by Yang et al. (2015), by differentiating
streams draining the interior of this low-relief region and those flowing around. Streams external (in green in Figure 9) have χ profiles mostly colinear with their main trunk stream - and more specifically with the local χ profile of their trunk stream, nearby their confluence. This suggests that these streams are to the first-order in equilibrium and keeping pace with the local incision of their trunk streams. In contrast, streams draining the Phobijka low-relief surface (in red in Figure 9) plot well above the Puna Tsang or Mangde Chhu in χ coordinates, with a high and low steepness downstream and upstream of a major





knickpoint, respectively. Such characteristics have been usually interpreted as reflecting disequilibrium with drainage area

gain by capture (Willett et al., 2014;Yang et al., 2015) (Figure 4c).

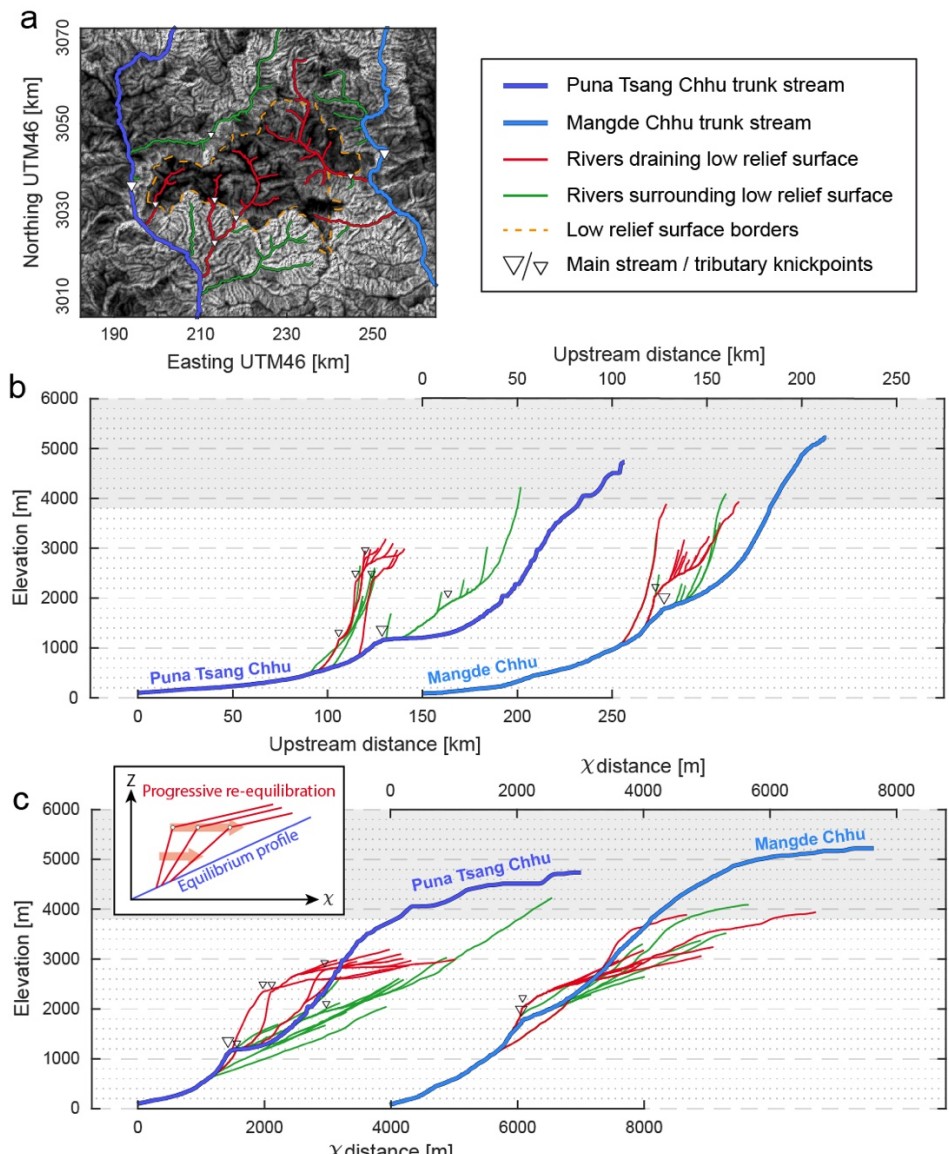

**Figure 9: Longitudinal and transformed river profiles around and within the Phobijka high-altitude low-relief surface**, following
the approach of Yang et al. (2015). Comparable results are verified for the Yarab surface (Figure S6 in supplementary materials).

a) Map of topographic relief of the Phobijka area (see Figure 1c for location), with a moving window of 500 m (scale bar as in Figure 10);
the identified low-relief region is delimited by the orange dashed line. Major knickpoints are reported with inversed triangular symbols.
Tributaries of the Puna Tsang (west) and Mangde (east) Chhu trunk streams are color-coded, according to whether they drain the low-relief
area (red) or are external to it (green).



b) Longitudinal river profiles of the Puna Tsang and Mangde Chhu, with main tributaries draining (red) or external (green) to the low-relief Phobijka area.

c) Transformed river profiles of the same rivers. Tributaries draining the low-relief area (red) plot above the main trunk streams as is observed in the case of area gain by river capture (Figure 4c). None of these profiles are concordant, indicating that these captures are not coeval and do not follow a coherent wave of incision propagating upstream. It should be noted that the higher the χ position of the knickpoints of the 580 internal streams, the lower the steepness of the pirate streams downstream of their knickpoint. This is particularly illustrated by the Puna Tsang Chhu tributaries and is interpreted as reflecting progressive re-equilibration of pirate streams toward an equilibrium profile after capture (inset). Tributaries external to the low-relief area (green) are locally colinear to the first-order with their main trunk stream, even though they gain drainage area by progressive divide migration (Figure 10).


Interestingly, the streams draining the Phobijka low-relief region have major knickpoints that are not concordant with the major knickpoint of their trunk stream in χ coordinates (Figure 9c). These streams show a peculiar pattern, most obvious in the case of the tributaries of the Puna Tsang Chhu : the greater the χ coordinate of the knickpoint, the lower the steepness of the stream segment downstream of the knickpoint, and therefore the closer (in χ coordinates) the stream profile gets to the 590 regional average set by its trunk river (Figure 9c). This suggests that the various portions of the high-altitude low-relief Phobijka area have been captured at various discrete and non-coeval times by these streams, and that some have had time to progressively partially re-equilibrate with their main trunk stream.

None of the streams have χ profiles clearly below the regional average driven by their main trunk channels (Figure 9c), as expected in the case of drainage area loss (according to Willett et al. (2014) and Yang et al. (2015)) (Figure 4c). It is 595 also noteworthy that the χ profiles of the streams external to the Phobijka surface (in green in Figure 9) remain close to - and not above - the regional average imposed by their main trunk river (Figure 9), even though they keep gaining drainage area by regressive erosion of the low-relief area (Figures 10).

Indeed, across-divide contrasts in χ, relief and elevation suggest that the Phobijka region is being regressively eroded all around by surrounding streams (Figures 10-11), in particular along its western divide with the Puna Tsang Chhu as contrasts 600 in χ and mostly elevation are highest. Contrary to the previous observations on low-relief regions drained by the Wang and Chamkhar Chhu rivers (Figures 7 and S3), there is no evidence here for a counteracting drainage area expansion of the streams draining this low-relief region downstream of it (Figure 10). The area of this low-relief region is shrinking and the streams draining it are losing drainage area. Based on both our observations and deductions from χ profiles (Figure 9c) with those from Gilbert metrics (Figure 10) of the various streams draining inside and outside the Phobijka area, we propose that the discrete 605 captures of this low-relief area has led to the punctual dramatic increase of drainage area of the pirate streams, but has not impeded the ongoing continuous regressive erosion of this elevated low-relief area - therefore accompanied by area loss - along its divide with other surrounding aggressor streams.





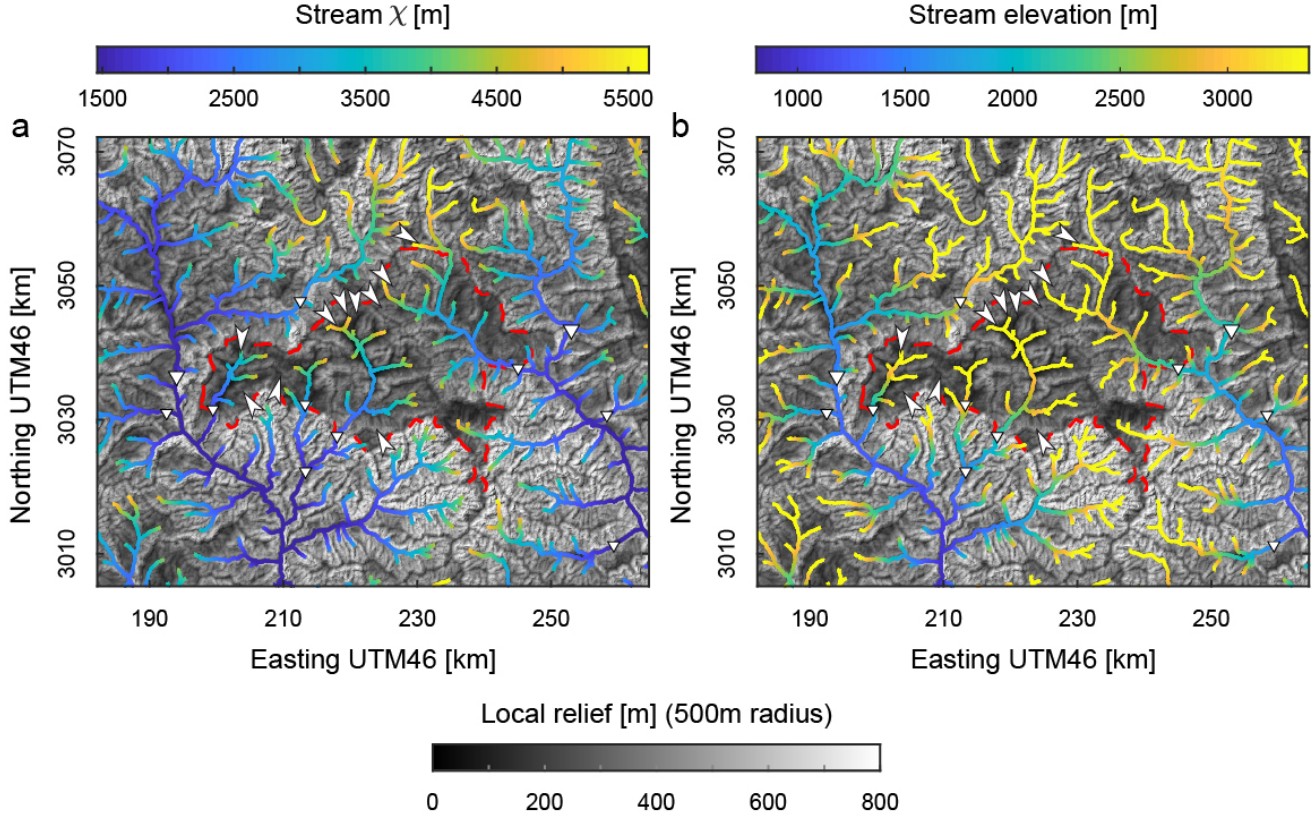

610

**Figure 10: Assessing divide mobility nearby the Phobijka high-altitude low-relief region** (see location on Figure 1c). Map represents local relief, with a moving window of 500 m, and the identified low-relief region is delimited by the red dashed line. Major knickpoints are reported with inversed triangular symbols and arrows locate the existence of morphological observations of regressive erosion (wind gaps), together with the direction of the associated divide migration. Similar observations are reported for the Yarab region (Figure S5 in supplementary material).

a) χ along the river network as a criterium to assess divide mobility (following (Willett et al., 2014)). The values of χ are determined for a concavity of 0.5.

b) Stream elevation as a criterium to assess divide mobility (following (Forte and Whipple, 2018;Whipple et al., 2017b)).

620





**Figure 11** *(previous page)*: **Satellite and field observations on the morphological contrasts across and around the Phobijka surface** (location on Figure 1c). Central map reports relief and χ along the river network as in Figure 10a. Satellite observations are taken from the Google Earth database.

a) ©Google Earth view (Image from CNES - Airbus and Maxar Technologies 2020) of the relief and elevation contrasts across the northwestern drainage divide of the Phobijka surface. White arrows locate observed beheaded channels (wind gaps), and indicate the direction of deduced drainage divide migration.

b) Field picture illustrating the high relief and steep slopes along the Dang Chhu (tributary of the Puna Tsang Chhu), along the northern boundary of the Phobijka surface. Note the contrast in relief and slope with picture d), across the local delimitation of the Phobijka surface.

c) ©Google Earth view (Image from CNES - Airbus 2020) of the relief and elevation contrasts across the southern limit of the Phobijka surface. White arrows locate observed beheaded channels, and indicate the direction of deduced drainage divide migration.

d) Field picture looking east of the Pele La pass (northern part of the Phobijka surface), where a filled valley is surrounded by higher relief rims. Note the contrast in relief with picture b), across the local delimitation of the Phobijka surface.

Similar observations and conclusions are reached for the streams draining inside and outside the Yarab low-relief region, located in eastern Bhutan and surrounded by the Kuri and Dangme Chhu (see Figures S5-S6 in supplementary material).

## 4.5 Results: general conclusions

Together, our results and field observations suggest that the morphological characteristics of the hinterland of Bhutan is not generated by a general wave of incision propagating upstream the drainage network. Indeed, all river profiles and their knickpoints are discordant in χ transformed coordinates (Figures 5c, 6 and 9 - by comparison to Figure 4b), whatever the spatial scale of investigation. This finding is supported by the secondary drainage network of rivers flowing through or around high-altitude low-relief regions since all these streams share a common base level and their χ profiles can be robustly compared (Figure 9). The documented χ profiles of major or secondary rivers evidence the existence of ubiquitous discrete and non-coeval river captures of low-relief regions in Bhutan.

Our results also suggest that a steady state morphology is far from being reached in the Bhutan Himalaya. Only few major rivers (the Kuri and Dangme Chhu, and to some extent the Puna Tsang and Mangde Chhu) have profiles associated with a potential equilibrium with tectonic and climatic forcing throughout the mountain range. The other large Himalayan rivers (the Chamkhar and Wang Chhu) show a peculiar dynamics on either side of their major knickpoint, with apparent drainage area contraction and expansion upstream and downstream of the knickpoint, respectively (Figure 7 and S3). This dynamic response of rivers is more pronounced at smaller spatial scales, where some secondary streams have captured high-altitude low-relief regions, and where others are progressively gaining drainage area by the regressive erosion of these same regions (Figures 9-11).



## 5 Discussion

### 5.1 Limits on our analysis and results

The comparison between the various Himalayan rivers throughout Bhutan whatever the dimensions of their drainage basins may be limited by lateral variations in boundary and forcing conditions. The widespread occurrence of Greater Himalayan series (Figure 1a) limits large-scale lithological variations. Tethyan or Lesser Himalayan series are also present in Central and Eastern Bhutan. However, spatial variations in bedrock erodibility associated with these variations is expected to be limited (Lavé and Avouac, 2001). Climate is everywhere tropical and wet. Precipitation varies mostly along a latitudinal transect across the range (Baillie and Norbu, 2004;Bookhagen and Burbank, 2006), with minor longitudinal variations from
east to west related to the rainshadow formed by the Shillong Plateau in the foreland (Grujic et al., 2006).

The Amo and Wang Chhu are the two rivers of westernmost Bhutan where significant lateral structural and tectonic variations may be expected, as they are located in the vicinity of the Paro structural window and of the Yadong graben (e.g. (Gansser, 1983;Greenwood et al., 2016;Long et al., 2011a)) (Figure 1a). Similarly, a structural window into Lesser Himalayan series prevails along the Kuri Chhu (Figure 1a), where variations in the seismic coupling of the MHT have been recently
evidenced (Marechal et al., 2016), suggesting other possible lateral variations in tectonics further east. In Western and Eastern Bhutan, we therefore cannot exclude the possibility of a tectonic forcing different from that encountered in Central Bhutan.

It should be noticed that the profiles and characteristics of the Wang and Amo Chhu in western Bhutan are comparable to the first order to those of the Chamkhar Chhu in Central Bhutan. Additionally, the Yarab high-altitude low-relief surface in Eastern Bhutan shares common attributes with the Phobijka low-relief patch in Central West Bhutan. In general, most
geomorphic features specific to the Bhutan Himalayas are found along a longitudinal band throughout this whole region of the Himalayas (Figure 1c). Together these observations suggest that the lateral variations in tectonics, climate or lithology documented in Bhutan have a relatively minor effect to the first-order on the observed morphology.

χ analyses are based on the stream power concept and do not include the effect of sediments on river incision (Gasparini et al., 2007). This can limit our interpretations for rivers with alluvial plains or for secondary streams capturing
low-relief regions. It has been shown that discordant χ profiles could be achieved for rivers enduring a common wave of incision when incision is dependent on the sediment flux (Giachetta and Willett, 2018). In this case, a positive correlation between drainage area and upstream knickpoint migration is expected as drainage area supposedly approximates available sediment fluxes (Giachetta and Willett, 2018), which we have not clearly found in our data (Figure S8 in supplementary material).

The comparison of river profiles in χ transformed coordinates requires a common base level for all streams. Tributaries with a common trunk stream meet this specific condition, but the comparison of major rivers with different and independent outlets is not straightforward. To overcome this limitation we assume that the base level of all major rivers is the Gangetic Plain, which has an almost constant elevation of c.a. 100-120 m south of the mountain front. For this reason, the



absence of a coherent wave of incision is mostly evidenced from the joint analysis of tributaries and of their trunk stream (Figure 9c), and only suggested by the comparison of major rivers (Figure 5b).

Some of our conclusions are therefore dependent on the classical limitations of χ analyses and of the stream power approach more generally speaking. However, these should not affect our main conclusions on river captures or migrating divides - and therefore on the unstable and dynamic pattern of the river network -, as they are further corroborated by field and satellite observations (Figures 8 and 11), as well as by across-divide contrasts in Gilbert metrics (Figures 7 and 10).


## 5.2 River captures, migrating divides and time scales of landscape response

Several transformed river profiles plot above the regional equilibrium profile of the largest rivers (Figure 5b), or above the profile of their trunk stream in the case of tributaries (Figures 6 and 9c). Such discordance in χ profiles is interpreted as reflecting an increase of drainage area due to river captures (Figure 4c). These captures can be dramatic for large river basins

such as for the Wang or Chamkhar Chhu, with high (> 1 km) major knickpoints (Figure 5c). They can be more modest for secondary streams draining high-altitude low-relief areas (Figure 9). In any case, our results suggest that river captures have occurred at all spatial scales in the hinterland of Bhutan. Discordant profiles and the observed variable position of knickpoints in χ coordinates in the case of secondary tributary streams favor the idea that these captures are discrete and not coeval in time, and that pirate streams are progressively returning to equilibrium (Figure 9c).

Secondary streams external to low-relief areas progressively gain drainage area by regressive erosion (Figures 10-11). Transformed profiles of these tributaries remain close to the regional trend of trunk channels (Figure 9c). This observation suggests that the time scale for channel re-equilibration is here faster than that of the perturbation driven by divide migration, consistent with conclusions from recent numerical experiments by Whipple et al. (2017b). This result implies that drainage re-organization by progressive divide migration is not expected to leave here a particular geomorphic signature, most probably

because of a relatively high erosional efficiency (Whipple et al., 2017b), in contrast with other field cases where the time for divide migration may have outpaced that of the erosional river response (Schwanghart and Scherler, 2020).

Only drainage area increase by discrete stream capture appears to leave here a clear imprint on the river network. These captures result in a large and abrupt drainage area increase, which requires a subsequent longer time for the rivers to adapt to their new boundary conditions. Such particular conditions are discussed by Whipple et al. (2017b) using the relative

ratio of the time scale of channel response to a characteristic recurrence time of large capture events. Following this reasoning, our results suggest that river captures leave an imprint on river profiles if captures occur more often than the time needed to adjust to their new conditions, or if the last capture event is younger than the time scale for channel response. These various considerations imply that only the most recent landscape evolution can be retrieved from morphometric approaches, leaving little - if no - possibility to access the morphologic conditions prior to the most recent captures.


## 5.3 Geomorphological characteristics of the Bhutan Himalaya, and possible interpretations



Our results were presented above following the various spatial scales of investigation. Below, we provide a synthesis of our observations for each geomorphic feature whatever associated spatial scale, and discuss possible interpretations of their origin. We finally compare our findings and inferences to previous interpretations.


### 5.3.1 Major knickpoints within the river network

Knickpoints along large Himalayan rivers of Bhutan separate two domains, with drainage area expansion downstream of the knickpoint and drainage area reduction upstream (Figures 7-8 and S3). Such dynamics on either side of a knickpoint has already been observed in other contexts of major river captures, such as for the Duero river in Spain and Portugal (Struth et al., 2019). Expansion of the drainage area downstream of the knickpoint leads locally to an increase in river incision and to the lowering of the base level, favoring and enhancing the upstream migration of the knickpoint (e.g. (Giachetta and Willett, 2018;Struth et al., 2019)). However, an additional feedback may operate as the drainage area reduction upstream of the knickpoint may limit and refrain the upstream knickpoint migration (Schwanghart and Scherler, 2020). Such feedback, by eventually significantly lowering the expected upstream migration of major knickpoints, may contribute to the preservation and surface uplift of the upstream low-relief region over time, to the over-steepening of the downstream river segments and therefore to the large dimensions of observed major knickpoints. If the reduction in the rate of knickpoint migration by upstream drainage area loss were significant, knickpoints along major Bhutanese rivers could appear as more or less stationary.

Major knickpoints are observed at all scales (Figure 6). They follow a specific spatial organization as most of them are localized in the Bhutan hinterland, nearby physiographic transition T2, whatever the size of the associated drainage basin (Figures 1b-c and 5a). In the case of non-coeval discrete stream captures at various spatial scales, this colocation would look surprising. It may therefore suggest that knickpoint migration remains limited in space, nearby T2. This indicates that these knickpoints originated and have possibly been maintained locally.

### 5.3.2 High-altitude low-relief regions

Transformed profiles of the rivers draining elevated low-relief regions all show a specific pattern indicative of river captures, whatever their spatial scale (Figures 5b and 9c). Conversely, no evidence for a counteracting drainage area loss by stream piracy appears in χ profiles of nearby rivers, as none of the profiles are below the regional equilibrium line delineated by the profiles of large Himalayan rivers (Figure 5b) or of their main trunk streams in the case of secondary low-relief regions (Figure 9c). It has been noticed that area loss by piracy is not as well expressed in transformed coordinates as is area gain by capture (see Figure 9b of Whipple et al. (2017b)). Because the absence of χ river profiles below estimated regional averages is observable at all spatial scales, even when dramatic captures are suspected, it could be postulated that such captures have not been at the expense of other rivers, but of the low-relief regions themselves if once isolated from the drainage network. Because the time scale for river response to changes in boundary conditions is expected to be relatively rapid (see section 5.2), the conditions that lead to the isolation of these landscape patches prior to their capture may not have left a morphologic record.





755        Local low denudation rates (Adams et al., 2016) and low relief (Figures 8 and 11), together with the presence of alluvial filling (Figures 3d-e) and stable soils (Figures 3f-g) (Adams et al., 2016;Baillie et al., 2004) in regions surrounded by higher-relief rims (Figures 3d-g, 8 and 11) indicate that the captured regions had most probably been already deprived of drainage area before capture. Discordant χ profiles at all scales (Figures 5b, 6b and 9c) suggest that these captures were most probably not coeval. The limits of low-relief regions most often coincide with drainage divides ("coincident divides" in the

sense of Willett (2017)) with numerous beheaded channels, in particular for secondary regions (Figures 8 and 11). These various observations and characteristics on local relief, valley filling, coincident divides and discordant transformed profiles favor the idea that these low-relief regions may have originated by area loss during in-situ dynamic re-organization of the drainage network (Willett et al., 2014;Yang et al., 2015), when following criteria recently proposed to discriminate such features from low-relief relict landscapes (Whipple et al., 2017a, 2016;Willett, 2017). The argument of relief inversion, with

filled valleys surrounded by higher-relief rims, is however to be taken with caution as such rims are expected to not be preserved during inward regressive erosion of low-relief regions (Willett, 2017).

       Alternatively, low-relief regions in the hinterland of Bhutan also meet some of the criteria proposed to indicate that they could be preserved relicts of pre-existing landscapes. Indeed, swath profiles across the Bhutanese hinterland illustrate well that these regions have similar altitudes and local relief to the first order (Figure 12), as if resulting from the surface uplift

and subsequent dissection of the same original topographic surface (after Whipple et al. (2017a)). However, this argument should also be taken with caution here. Indeed, morphometric analyses may mostly detect the lowest-relief patches, as these offer the highest relief contrast with other regions (Figure 1c). We cannot therefore discard the possibility that other low-relief patches (but with relatively higher relief) exist at other altitudes. In the case of low-relief regions formed by area loss during drainage re-organization, local relief is expected to correlate with altitude as relief reduction is concomitant to surface uplift

(Whipple et al., 2017a): in this case, comparable relief could be expected at similar altitudes. The similar altitude and local relief of low-relief regions (Figure 12) may therefore either favor the idea of relict landscapes or be an observational bias.

       In conclusion, current observations do not permit to clearly discriminate whether low-relief regions of Bhutan are remnants of former landscapes or whether they were formed dynamically in-situ, even though the ample evidence for dynamic network re-organization may favor the latter idea. In the former case, they could be used as geomorphic markers to retrieve

the surface uplift and the timing of the changes in tectonic and climatic boundary conditions that lead to their uplift and dissection; in the latter case, this information is not accessible.

       These interpretations, whether of relict landscapes or of relief inversion by area loss (or a mix of the two processes), only describe the type of morphological response of the river network leading to the formation of these features, but do not provide the processes driving this response. We emphasize that the high-altitude low-relief regions are all co-located in the

hinterland of Bhutan, in between physiographic transitions T2 and T3, whatever their spatial scales and dimensions (Figure 1c). This strongly suggests that they were most probably formed only locally and not pervasively throughout Bhutan, appealing for a local process driving and supporting their formation, preservation and subsequent dissection.


Earth **Surface**
**Dynamics**
Discussions

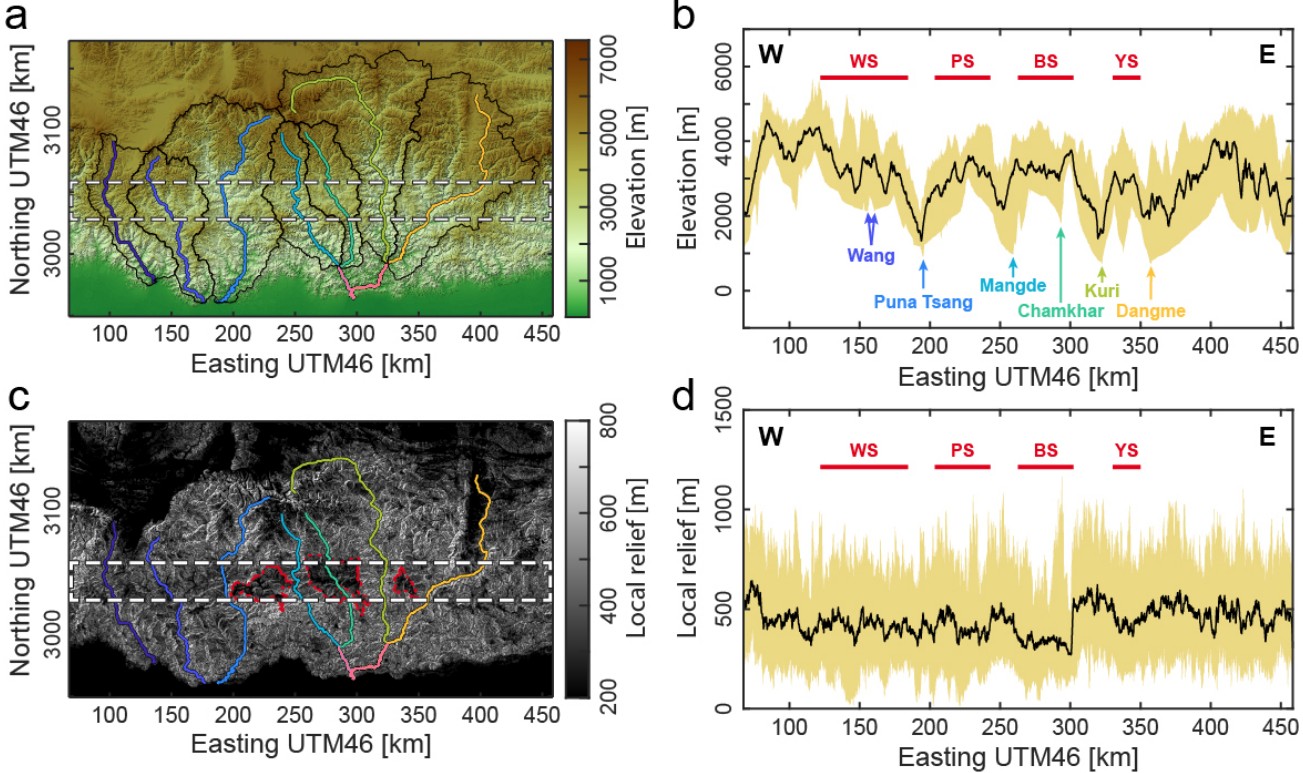


**Figure 12: Topography and relief along the hinterland of the Bhutan Himalaya, where low-relief surfaces are located.**

WS: Wang Surface; PS: Phobijka Surface; BS: Bumthang Surface; YS: Yarab surface.

a) Topographic map of the Bhutan Himalaya from ALOS World 3D – 30m (AW3D30) DEM data. Main drainage basins are delineated by black lines and associated main rivers are color-coded (color-code as in Figure 1). The dashed rectangle locates the swath profile represented in b).


b) Topographic swath profile along the Bhutan hinterland, with elevations taken within the band illustrated in Figure 10a. Location of major rivers and of low-relief regions are reported.

c) Map of local relief, as calculated from the topography shown in Figure 10a, with a moving window of 500 m. Major high-altitude low relief areas are delineated by dashed red lines. Main rivers are located and color-coded. The dashed rectangle locates the swath profile

represented in d) and encompasses the same region as that of Figure 10a.

d) Swath profile of relief along the Bhutan hinterland, with relief taken within the band located in Figure 10c. Low-relief regions of the mountain range interior are reported. The lower relief values of the swath profile are to be considered for low-relief regions.

**5.3.3 The spatial organization of all major knickpoints and low-relief regions calls for a common local tectonic origin**

Our investigations illustrate the highly dynamic - and eventually unstable - pattern of the river network in the Bhutan hinterland, with migrating divides and numerous non-coeval stream captures at various spatial scales. Despite this dynamics,



both major knickpoints and high-altitude low-relief regions, whatever their spatial scale, all co-locate in the hinterland of the Bhutan Himalaya, along an approximately west-east oriented spatial band nearby physiographic transition T2 or in between T2 and T3 (Figures 1b-c). As emphasized previously for each specific geomorphic feature, this observation calls for a common local genetic origin for all these features and for the observed network dynamics. These observations hold throughout most of the Himalayas of Bhutan, even in Eastern Bhutan where no major knickpoint exists along the Kuri and Dangme Chhu at these latitudes (Figures 1 and 5). Indeed, the presence of the Yarab low-relief region, localized in between these two major rivers (Figures 1c), indicates that a similar mechanism is also ongoing in this part of the Himalayan range (Figures S5-S6). The absence of a major knickpoint on adjacent large rivers reveals that these have sufficient power to adjust to local conditions. The only minor exception to the observed colocation lies in the low-relief surface and major knickpoint of the Wang Chhu in Western Bhutan, which are deported c.a. 20-30 km southward from the observed longitudinal band (Figures 1 and 5a).

This spatial geomorphological organization is not expected to be controlled lithologically as it is clearly not correlated to first-order surface geology (Figure 1a). Alternatively, the orientation of the swath containing all major knickpoints and low-relief regions supports the idea for a local tectonic driver in the mountain hinterland, as this orientation is parallel to expected active tectonic structures (such as the frontal MFT or MBT - Figure 1a) or associated geomorphic expressions (such as the High Himalayan range expectedly above a mid-crustal ramp of the MHT (Coutand et al., 2014;Diehl et al., 2017;Le Roux-Mallouf et al., 2015;Marechal et al., 2016;Singer et al., 2017)) (Figure 1b and 2).

Additionally, the east-west swath encompassing the investigated geomorphic peculiarities of the Bhutan Himalaya also approximately follows the orientation of the precipitation pattern in Bhutan, and is slightly north of a spatial band of highest precipitation rates (e.g. (Bookhagen and Burbank, 2006;Grujic et al., 2006)). Such climatic pattern results from the major orographic barrier formed by the steeply rising topography between physiographic transitions T1 and T2 (Figures 2a-b), for which tectonics is most probably the forcing driver as uplift is needed at some point to build up topography. As such, we propose that the peculiar geomorphic dynamics, the steep topography between T1 and T2 and the associated local climatic pattern all share a common initial tectonic driver. The observed and investigated landscape characteristics is therefore interpreted as reflecting to the first order active tectonics in the hinterland of Bhutan, but the interplay between local uplift, the resultant high slopes and climate may modulate and enhance local landscape dynamics.

### 5.3.4 Testing previous interpretations based on our findings

The absence of a coherent wave of incision moving upstream the drainage network (Figures 5b, 6b and 9c) and the spatial organization of observed geomorphic features along a longitudinal band, at all scales and only within the hinterland of the Bhutan Himalaya (Figure 1), exclude previous interpretations relying on any large-scale change in tectonics or climate over Bhutan. This is the case for the concept of tectonic rejuvenation of Bhutan, relative to the Himalaya of Central Nepal, proposed by Duncan et al. (2003), or for the general surface uplift of Bhutan consequent to decreasing precipitations in the rainshadow of the rising Shillong Plateau proposed by Grujic et al. (2006). The dimensions of the major knickpoints of some



of the large rivers, with heights of > 1km in the cases of the Amo, Wang or Chamkhar Chhu (Figures 2c and 5), are most probably too large to be related to an upstream migrating wave of incision due to a change in relative base level.

Baillie and Norbu (2004) proposed that the variety of river profiles were related to lateral variations in tectonic uplift throughout Bhutan. This may potentially explain discordant χ profiles of major and large rivers (Figure 5b). In this case the
spatial co-location of knickpoints throughout Bhutan, within a west-east swath (Figures 1 and 5a), would indicate that the location of active tectonics is somehow coherent along strike but that only the rates of active deformation are laterally variable. Even though lateral variations in the rates of uplift cannot be excluded, these are not expected to vary significantly to the first-order, and in particular from one valley to the other. Additionnally, the similar steepness of large rivers south of T2, located variably in western (Puna Tsang Chhu), central (Mangde Chhu) and eastern (Kuri and Dangme Chhu) Bhutan (Figure 5b) calls
for similar rates of uplift to the first-order, given the similar lithologic (Figure 1a) and climatic (e.g. (Bookhagen and Burbank, 2006;Grujic et al., 2006)) conditions. Also, the presence of the high-altitude low-relief Yarab surface in eastern Bhutan indicates that similar first-order conditions prevail in this part of the Himalaya even though the two major surrounding rivers (the Kuri and Dangme Chhu) have no major knickpoint. First-order similar tectonics, in terms of patterns and also most probably in terms of rates, are therefore expected throughout Bhutan despite the variability of fluvial profiles. In fact, this
variability is rather expected to relate to the various responses and capacities of the different rivers to adjust to local tectonics, as the largest rivers have limited or absent major knickpoints, and vice versa (Figure 3 and Table 1, Figure S8 in supplementary material).

Interestingly, Adams et al. (2016) proposed that major knickpoints and low-relief landscapes were the geomorphic response to recent uplift over a blind duplex in the Bhutan hinterland. This interpretation is seducing as it fits the idea that the
observed geomorphic features need to be primarily sustained locally by ongoing active uplift. Here, the proposed tectonic perturbation originates in the hinterland and not at the base level of major rivers, which may account to some extent for their discordant χ profiles (Figure 5b). In the details, however, their idea and subsequent inferences fail to account for some of our observations. In the case that knickpoints were removing fill deposits while migrating upstream and upwards, upstream knickpoint migration (upstream from the tectonic perturbation in the hinterland) would be expected to correlate with knickpoint
altitude, upstream drainage area and the extent of uplifted alluvial fill, which is not specifically observed (Figure S8). Indeed, the co-location of all major knickpoints whatever considered spatial scale, from major to local rivers (Figure 1), indicates that these knickpoints are most probably relatively stationary or migrating only locally (see section 5.3.1). Whether low-relief alluvial valleys - either along major rivers or in the upstream portions of secondary reaches - are relics of local landscapes prior to initiation of uplift in the hinterland or whether they are formed dynamically in-situ by the re-organization of the river
network is not straightforward to elucidate (see section 5.3.2). For this reason, the derivation of cumulated tectonic uplift from a theoretical river profile and the present-day profile in which the upstream portions are perturbed by the locally rising base level may not be appropriate. Also, the use of denudation rates to assess the timing of initiation of uplift is questionable as these rates may not be representative of actual uplift rates when derived from local catchments that are either victims or aggressors as their drainage divides are continuously migrating inwards or outwards, at the benefit or expense of nearby



catchments, respectively (e.g. (Sassolas-Serrayet et al., 2019)). It should be also emphasized that uplift rates are expected to vary across strike, with higher rates over the blind ramp and with lower rates downstream and upstream of it, further challenging the comparison between denudation rates measured in high-relief canyons and those in upstream alluvial or isolated low-relief valleys. As a final point, the idea of Adams et al. (2016) of uplift over a blind ramp fits our conclusions that the observed geomorphic characteristics needs to be sustained locally by tectonic uplift, even though in the details some of

their deductions do not account for the observed high dynamics and instability of the river network.

**6 Conclusion**

Based on field observations and on a quantitative and qualitative morphometric analysis of the mountain hinterland in Bhutan at various spatial scales, we have further documented the landscape dynamics of this part of the Himalayas where

out-of-equilibrium morphologies have been described. We find that the various geomorphic features of the Bhutanese landscape, such as major knickpoints or high-altitude low-relief areas, are not related to the migration of a wave of incision upstream the river network, as expected from most previous interpretations. Rather, our analysis emphasizes the existence of numerous drainage captures, at various spatial scales, spatially discrete and temporally non-coeval, with partial re-equilibrations of captured drainages in some cases. In addition to these captures, drainage divides are found to migrate in

various parts of the hinterland, in particular around or downstream elevated low-relief areas, with efficient re-equilibration of river profiles as expected in the case of relatively fast landscape response times.

Our results therefore emphasize the existence of an unstable and dynamic landscape, which however obeys a specific spatial organization along a longitudinal spatial band in the mountain range interior. This latter key observation supports the idea that the documented geomorphic dynamics needs to be sustained locally by tectonic uplift. Even though the idea proposed

by Adams et al. (2016) of uplift over a blind ramp in the hinterland fits our first-order conclusions, some of their deductions do not capture or account for the observed high dynamics of the river network. This is most probably because they base their deductions on the comparison to a landscape evolution simulation in which drainage divides remain stationary. Future work exploring the initial idea of Adams et al. (2016) of active tectonic uplift in the mountain hinterland but with account on the possible dynamic response of the river network, with pervasive migrating divides and river captures, is therefore needed.

Finally, because the drainage network is not stable with ubiquitous divide migrations in the hinterland, the previous use of denudation rates as a proxy for tectonic uplift rates in Bhutan (Adams et al., 2016;Le Roux-Mallouf et al., 2015) may be problematic, and results should be re-evaluated. Our work, therefore emphasizes the need for a precise investigation of landscape dynamics and disequilibrium over various spatial scales as a first essential step in morpho-tectonic studies of active landscapes.




**Supplement material**

Additional figures are provided as supplementary material.

**Author Contribution**

MS designed and conceptualize the research objectives of this study. She also wrote the main draft of the manuscript with contributions from all co-authors. TSS carried the presented geomorphic analyses and prepared all associated figures. RC gathered all the necessary financial support that lead to the work presented here. All co-authors participated to field work, scientific discussions and the co-editing of the manuscript.


**Competing interests**

The authors declare that they have no conflict of interest

**Acknowledgements**

The fruitful discussions that led to this manuscript rose during field work in Bhutan, and our friend and driver Phajo Kinley (Department of Geology and Mines, Thimphu) is warmly thanked for taking good care of us all during our various journeys. Through the work of MS, this study stands as IPGP contribution # XX.

**Financial support**

This study benefitted from grants from the Agence National de la Recherche (France) attributed to RC, through projects ANR BhutaNepal (ANR grant # ANR-13-BS06-0006) and TopoExtreme (ANR grant # ANR-18-CE01-0017). TSS benefitted from a PhD grant attributed by the French Ministry of Higher Education and Research. Through the work of MS, this study contributes to the IdEx Université de Paris ANR-18-IDEX-0001.

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
