# Peer review of "Topographic disequilibrium, landscape dynamics and active tectonics: an example from the Bhutan Himalayas."

_Earth Surface Dynamics, 2020_

## Referee Comment (RC1) · Anonymous Referee #1 · 14 Jan 2021

In the submitted paper "Topographic disequilibrium, landscape dynamics, and active tectonics: an example from the Bhutan Himalaya", Simoes et al perform a topographic analysis of the Bhutanese Himalaya, with a special focus on the low-relief, high-elevation surfaces that have attracted prior attention in this portion of the range. They primarily approach this landscape from the perspective of evaluating drainage divide instability and the extent to which this either complicates prior results or helps to demonstrate what may be driving the landscape form in this region.

Technically the paper is fine, the analyses are appropriate and the individual interpretations that flow from these analyses are mostly warranted and/or logical. My main

issue is that the motivation of the paper, and their addressing of this motivation in the discussion/conclusion, seems a bit problematic. They set up the paper by describing the landscape, its interesting morphology, and some of the prior tectonic and geomorphic interpretations. The problem is that they end up misrepresenting the interpretation from Adams et al, 2016 in the motivation and then after their analysis essentially confirm most of what Adams was arguing for, but still indicating that it wasn't what Adams was arguing for (e.g. they suggest that the Adams model was inconsistent with nearly static knickpoints, where in the Adams model explicitly argued for nearly static knickpoints). Similarly, they describe the Adams paper as arguing for the preservation of a "relict- landscape", where in detail the Adams paper explicitly argues against a relict landscape preservation hypothesis (some of this may be semantic, i.e. it seems like they are using an odd, non-standard definition of relict landscapes which differs from what is normally used, so this could be fixed by clarifying what they mean by specific terms).

There is certainly value in documenting some of the interesting and nuanced drainage network reorganizations that are occurring in this landscape, but the paper suffers from seeming to set up sort of a false controversy (and it is unfair to the Adams paper in that ultimately, most of the observations here confirm, or are consistent with, hypotheses put forward in the Adams paper). I think recasting the introduction / conclusion of the paper to be less about testing or addressing a controversy and more about exploring another interesting aspect of this landscape that wasn't really addressed in the prior work by Adams et al (various years), i.e. drainage network instability, and thinking about how this is being driven / influenced by the tectonic context seems much more appropriate. Ultimately, coming at it from this approach may allow for more interesting and meaningful interpretations and/or implications.

Line-by-line comments:

L70-71: This statement at least does not reflect one of your cited references, i.e. Adams et al, 2016 argue for in-situ development of the low-relief surfaces from blind

duplexing, which they argue may be structurally linked to the development of the Shillong plateau, but definitely is not representative of "relicts of former climatic or tectonic conditions".

L118: The Gilbert metrics are formally defined in Forte & Whipple, 2018, not in the Whipple et al, 2017 JGR-ES paper.

L145-144: You might also consider citing the recent Adams et al, 2020 (Adams, B.A., Whipple, K.X., Forte, A.M., Heimsath, A.M., Hodges, K.V., 2020. Climate controls on erosion in tectonically active landscapes. Science Advances 6. https://doi.org/10.1126/sciadv.aaz3166) as their analysis of this region is also consistent with a relative invariant erodibility for much of the Bhutan region.

L281-282: "low-relief hanging fill valleys can be interpreted as relict landscapes formed locally", this seems like a very odd way to describe a landscape, that in the interpretation you're describing, is actively maintained by uplift of blind duplexes and the original authors describe as forming in-situ and explicitly reject the idea of these being "relict landscapes" in the traditional sense. I would consider rewording this to avoid confusion.

L293-294: See previous point, i.e. at least when considering the Adams model, they explicitly reject the idea of these being relict landscapes, at least in the way this term is typically used (e.g. the Whipple et al – Willett et al paper/comment and reply chain that you cite). I think you either need to reword this and other places or be much more explicit about how you are using/defining relict landscape, because this seems to be a non-standard way of describing them and it is (1) confusing and (2) misrepresents the results of previous work if you apply the more standard definition of relict landscapes.

L406: Yes, but for a relatively short time, this is one of the key points of Whipple et al, 2017 (JGR-ES) paper.

L412: As earlier, Forte & Whipple, 2018 is the more appropriate reference here as this paper highlights the complications of base level choice.
L663-665: It would be useful perhaps to consider this in the context of the aforementioned Adams et al, 2020 paper. I.e. they demonstrate that the large magnitude variations in precipitation rate have an important control on the scale of the topography (ksn) and its relation to erosion rates. The Gilbert metrics shouldn't be influenced by this, but you've calculated chi assuming static K and precipitation (as most do), but in the context of the Adams result, I wonder if calculating chi with the modern spatially variable precipitation would alter the chi patterns? My hunch would be no, and I don't necessarily think you need to demonstrate this, but I think it would be good to acknowledge that there are pretty significant precipitation gradients and they have been shown to influence topography and the reflection of erosion rates within topography.

L745-766: A fundamental problem with applying the Yang et al hypothesis and/or the Willett criteria for recognizing area loss/gain in chi-transformed river profiles to this landscape is the hypothesized presence of relatively discrete structural breaks (i.e. the blind duplex of Adams). This fundamentally violates some of the underlying assumptions in a pretty big way. More specifically, in chi-transformed space, a river profile responding to a growing duplex is going to look like a river having gained area. The key as you allude to elsewhere is the spatial consistency of the pattern, and thus probably not all of the area gain signatures are tectonic related, but some might be. You ultimately exercise caution in terms of applying the area loss feedback mechanism, which is warranted, but I think a more nuanced look at what you might expect in a structurally complex setting like this is important.

L865-867: This is confusing, as Adams et al, 2016 explicitly argues for the knickpoints generated by the duplex to be fixed in longitudinal position (e.g. their figure 6 or figure 10), which seems consistent with your observations, but you cast it as though this is an observation that disagrees with the Adams et al, 2016 model?

L868: See prior comments about the confusing use of relict- landscapes.

L885-887: As noted previously, this seems to at least misrepresent the conclusions of

some of the prior work.

---

## Referee Comment (RC2) · Wolfgang Schwanghart (Referee) · 5 Feb 2021

I enjoyed reading the manuscript by Simoes et al. It summarizes the controversies around the enigmatic high-elevation low-relief landscapes in Bhutan. Based on geomorphometric analysis of river profiles and drainage divides, the authors emphasize the role of divide migration in shaping the low-relief regions and conclude that existing denudation rates should be reevaluated given these dynamics.

Overall, the manuscript is well written, although lengthy at times. Figures have a high quality, but could be simplified and better annotated for better readability (see comment below). The number of figures seems adequate, but some of the plots appear in very

similar form twice (for example Fig. 7a and the map in Fig. 8). This could be avoided. The methods are sound and described in a way that they are reproducible. In parts, the results are intermingled with interpretations which would be better placed in the discussion (e.g. 502-506).

As reviewer #1 notes, I also find it difficult to see how the results of this study corroborate or contradict the findings of the studies by Adams et al. Moreover, I find it difficult to follow why other concepts of tectonic rejuvenation (Duncan et al. 2003) are dismissed, based on the grounds that there is an absence of a coeherent wave of incision. Shouldn't it be expected that such a coherent wave is missing given that drainage divide mobility may be a process that prevails throughout this landscape?

The observation that catchments downstream of knickpoints are expanding is intriguing, but the mechanism that generates the expansion remains unclear. The studies by Struth et al. (2019) and Giachetta and Willett (2018) are referenced in this context, but these studies show examples where expansion happens downstream of areas with internal drainage and that were integrated in the flow network. Are endorheic basins a possible explanation for the preservation of these landscapes? And if not (which is quite likely given the humid climate), what could be an alternative intepretation? An hypothesis that might be brought forward could be the availability of sediments mobilized from the alluvial plains upstream that would act as tools accelerating incision downstream which would propagate towards the divides.

I find it difficult to read some of the figures. The combination of a grayscale depiction of topographic relief (which is quite printer-unfriendly), and colored networks makes some maps really busy and difficult to read. The colored stream networks (e.g. in Fig. 2c and 5) have variations in blue and green that are quite subtle or not resolved by my printer. Consider to label the river profiles in the plots rather then using a legend.

In addition to above major comments, I have numerous minor comments listed below:

29: Remove "indeed". In general, the text contains numerous filler words, which could

be avoided.

35: Remove "first-order". I have seen this term a couple of times in this manuscript, but I don't know what it actually means in most contexts. For example, in line 63, I don't understand the term "first-order consistency".

253: the term "rather relative" is quite vague, as is the term "rather similar" in line 256.

394: remove 'long-distance'

395: migrate upstream in response

395: what do you mean by 'common process'.

396: perhaps rephrase "are expected to cluster in transformed coordinates".

411: , however,

419: Consider shortening this sentence: These complementary methods enable a more careful assessment of divide migration direction and drainage network reorganization.

424: Perhaps rephrase: Based on visual interpretation of longitudinal and chi profiles, we identify three profile types of major rivers in Bhutan.

425: Avoid the term 'simple'. Rather write that these profiles are concave upward with no remarkable knickpoint.

426: Remove 'rivers like'

433: 'intermediate characteristics' is a bit vague.

441: above 3800 m

456: Not sure what "better organized" means

459: Remove 'clearly' twice

462: Remove 'first-order'

465: You may better write "analyze the geometry of". The dynamics will be inferred from the geometry.

476: The sentence is vague: rivers compare well and are colinear to the very first-order. I am also not sure what you mean by 'first order' as used in the next sentence. In addition, this part mixes observations (or results) and interpretation.

479: On which basis do you judge that a knickpoint chi-value is discordant from another. Consider providing quantitative evidence. One possible way to report these differences in chi values could involve calculating the necessary change in area required so that the locations of knickpoints are the same in chi space. This would allow readers to appreciate the differences in knickpoint locations and would provide a way to eventually exclude or consider divide dynamics as potential mechanism that creates the differences in knickpoint locations.

497: remove words like "clearly"

502: this paragraph should be better placed in the discussion

549: Avoid the term "dramatic" (which is found several times in the manuscript).

550: While expansion is the right term, I don't like the term contraction is this context, because it implies that there are processes that exert a stringent force. I would rather use "specific pattern of drainage area loss and expansion".

565: Better place this sentence in the discussion.

640: Such summaries are generally helpful. However, you may consider moving it to the beginning of the discussion, also.

645: robustly? Robust in statistics usually means insensitive to outliers. I am not sure what it means here.

659: remove "whatever the dimensions of their drainage basins"

691: replace "the classical" with "known"

692: replace "more generally speaking" with "in general"

699: rephrase to avoid "dramatic"

756: what are "stable soils"? There is no Fig. 3g.

835: This assertion of an "absence of a coherent wave" needs better quantitative justification, as mentioned above. And given that divide dynamics are an important process, isn't that what you would expect irrespective of the absence or presence of a large-scale tectonic or climate signal?

---

## Author Comment (AC1) · 4 Mar 2021

NB: Hereafter, all comments posted by Anonymous Referee #1 (RC1) are preceded by "RC1", and are followed by the authors' response (preceded by =>).

RC1: In the submitted paper "Topographic disequilibrium, landscape dynamics, and active tectonics: an example from the Bhutan Himalayas", Simoes et al perform a topographic analysis of the Bhutanese Himalaya, with a special focus on the low-relief, high-elevation surfaces that have attracted prior attention in this portion of the range. They primarily approach this landscape from the perspective of evaluating drainage divide instability and the extent to which this either complicates prior results or helps to

demonstrate what may be driving the landscape form in this region. Technically the paper is fine, the analyses are appropriate and the individual interpretations that flow from these analyses are mostly warranted and/or logical. My main issue is that the motivation of the paper, and their addressing of this motivation in the discussion/conclusion, seems a bit problematic. They set up the paper by describing the landscape, its interesting morphology, and some of the prior tectonic and geomorphic interpretations. The problem is that they end up misrepresenting the interpretation from Adams et al, 2016 in the motivation and then after their analysis essentially confirm most of what Adams was arguing for, but still indicating that it wasn't what Adams was arguing for (e.g. they suggest that the Adams model was inconsistent with nearly static knickpoints, where in the Adams model explicitly argued for nearly static knickpoints). Similarly, they describe the Adams paper as arguing for the preservation of a "relict-landscape", where in detail the Adams paper explicitly argues against a relict landscape preservation hypothesis (some of this may be semantic, i.e. it seems like they are using an odd, non-standard definition of relict landscapes which differs from what is normally used. so this could be fixed by clarifying what they mean by specific terms).

=> We do sincerely thank RC1 for appreciating the quality of our analyses and of our work on the dynamics of the river network in the hinterland of the Bhutan Himalaya. We do regret the misunderstanding on how we decribe how Adams et al (2016) meet our final conclusions on the fact that active uplift is the most probable supporting mechanism for the observed peculiar morphologies, even though in the details some of their model inferences deviate from our observations. The work by Adams et al (2016) appears as the most advanced interpretation of these peculiar morphologies and we wish to give good credit to their work. By emphasizing the differences between their model and our observations, we wish to point the way to move forward in the future in the modeling of the observed morphologies - and not to discredit their work. We will take good care to modify the text so as to make sure this is clearly stated in the revised version of the manuscript.

**ESurfD**
Additionally, in the details:

- static versus migrating knickpoints: we disagree with RC1. Indeed, Adams et al (2016) suggest that major knickpoints migrate upstream. This is repeatedly stated in their manuscript that we have carefully re-read (examples are provided below), and is illustrated in their figures 6 and 11. Indeed in figure 6 of Adams et al 2016, the knickpoint migrates from position ~90 km in b) to position ~110 km in c) while the model evolves in time; in figure 11 of Adams et al 2016, the knickpoint migrates upstream from positions ~12 km (river 2) and ~15 km (river 1) to positions ~15 km (river 2) and ~18 km (river 1), repectively. This migration remains limited - even though the time that separates each of these model snapshots is not reported - , but goes together with the idea stated in the manuscript that knickpoints migrate ustream in their model.

Citing some examples of Adams et al 2016 for what concerns migrating vs. static knickpoints, just by searching for the words "migrating knickpoint" throughout the text (pages refer to the PDF): "Failure to match the rising local base- level set by the migrating knickpoints with a similar deposition rate would have led to a defeated, ponded river and an internally drained basin " (p15); " The stippled pattern marks the packages of sediment accumulating upstream of a migrating convex knickpoint (black dot) and forming the migrating concave knickpoint upstream (white dot) " (caption of Figure 6); " Comparisons with our landscape evolution model and the observed sediment deposits both suggest that the low-relief landscapes of Bhutan were actively aggrading as they adjusted to the local baselevel rise created by a migrating convex knickpoint " and " I is the incision rate into bedrock at the position of the migrating convex knick- point"(p17); " Our landscape evolution experiment also supports the hypothesis that such low- relief landscapes are transient features whose positions are controlled by head- ward migrating, convex knickpoints, as evident from the dichotomy in erosion rates between the low-relief landscapes and adjacent canyons. " (p. 23).

- relict landscapes: we agree with RC1 that our initial terminology and phrasing may have been confusing and misinterpreted. By "relict landscapes", we referred here to

**ESurfD**
the fact that former valleys of the mountain hinterland had been preserved (even though subsequently filled with sediments) and uplifted in Adams' model. This lead Adams et al 2016 to use the uplifted position of these alluvial valleys as a marker of uplift above a theoretical initial river profile. In fact, in their model, the overall shape of the valleys are remnants of former incisional valleys (explaining that we used the term "relict" for 'remnant', initially), and that only alluvial filling occurred in-situ during uplift (what Adams et al 2016 termed 'in situ formation of the valleys' - probably also confusing). We recognize that the term "relict landscape", classically used in the case of landscapes formed along mountain foothills and grading to the foreland, is not adapted here. We will take care in our revision to correct for this, and will rather refer in the case of the Adams' model to remnants of formerly incising valleys, subsequently uplifted and filled with sediments.

RC1: There is certainly value in documenting some of the interesting and nuanced drainage network reorganizations that are occurring in this landscape, but the paper suffers from seeming to set up sort of a false controversy (and it is unfair to the Adams paper in that ultimately, most of the observations here confirm, or are consistent with, hypotheses put forward in the Adams paper). I think recasting the introduction / conclusion of the paper to be less about testing or addressing a controversy and more about exploring another interesting aspect of this landscape that wasn't really addressed in the prior work by Adams et al (various years), i.e. drainage network instability, and thinking about how this is being driven / influenced by the tectonic context seems much more appropriate. Ultimately, coming at it from this approach may allow for more interesting and meaningful interpretations and/or implications.

=> We regret that our work has been seen as setting any kind of controversy, as our objectives were not those. Indeed, we aimed at documenting and understanding the dynamics of the river network and the relative time scales for landscape response from the particular example of the Bhutan Himalaya where out-of-equilibrium morphologies have been documented. Our study also provides an interesting field example where the

**ESurfD**
classical use of morphology to derive rates of active tectonics is to be done with great caution. When comparing our results to previous work and interpretations, we wish to give good credit to all previous work, and in particular to the work by Adams et al (2016) that proposed up to now the best model that fits most of our observations. Even though we agree with Adams et al 2016 on the idea of active uplift in the mountain hinterland, our results emphasize the limits of their model, and by doing so aims at pointing out future directions of work, in particular by proposing to better include the dynamics of the drainage network when modeling the lansdcape response to active tectonics (see our conclusion). We will take good care when revising the manuscript to clarify this so as to keep fair with this pioneering work.

RC1: L70-71: This statement at least does not reflect one of your cited references, i.e. Adams et al, 2016 argue for in-situ development of the low-relief surfaces from blind duplexing, which they argue may be structurally linked to the development of the Shillong plateau, but definitely is not representative of "relicts of former climatic or tectonic conditions".

=> As stated above, we agree that using the word "relict" in the case of the Adams' model may be confusing and should be avoided.

RC1: L118: The Gilbert metrics are formally defined in Forte & Whipple, 2018, not in the Whipple et al, 2017 JGR-ES paper.

=> We do not fully agree with RC1. The idea of Gilbert metrics was first proposed in Whipple et al 2017 JGR Earth Surface (see for instance section 5 of this 2017 manuscript "Topographic Metrics for Recognizing Mobile Divide", p 263-265 ; in addition to their section 7.2. "Utility of Topographic Metrics of Erosion and Divide Mobility", p 269-270) - even though these metrics were not initially termed "Gilbert metrics". These metrics were named as such, further expanded and discussed in the Forte and Whipple 2018 paper, and we agree that this manuscript should be also cited here.

RC1: L145-144: You might also consider citing the recent Adams et al, 2020
(Adams, B.A., Whipple, K.X., Forte, A.M., Heimsath, A.M., Hodges, K.V., 2020. Climate controls on erosion in tectonically active landscapes. Science Advances 6. https://doi.org/10.1126/sciadv.aaz3166) as their analysis of this region is also consistent with a relative invariant erodibility for much of the Bhutan region.

=> We thank RC1 for this suggestion, which will be integrated in the revised manuscript.

RC1:L281-282: "low-relief hanging fill valleys can be interpreted as relict landscapes formed locally", this seems like a very odd way to describe a landscape, that in the interpretation you're describing, is actively maintained by uplift of blind duplexes and the original authors describe as forming in-situ and explicitly reject the idea of these being "relict landscapes" in the traditional sense. I would consider rewording this to avoid confusion.

=> As stated and explained in detail above, we recognize that the term "relict" was confusing and not used following the classical meaning. This sentence will be rephrased.

RC1:L293-294: See previous point, i.e. at least when considering the Adams model, they explicitly reject the idea of these being relict landscapes, at least in the way this term is typically used (e.g. the Whipple et al – Willett et al paper/comment and reply chain that you cite). I think you either need to reword this and other places or be much more explicit about how you are using/defining relict landscape, because this seems to be a non-standard way of describing them and it is (1) confusing and (2) misrepresents the results of previous work if you apply the more standard definition of relict landscapes.

=> See previous answers above. Indeed, we agree that "relict landscape" was not meant in our manuscript in the classical way, but rather in the sense that alluvial valleys are interpreted as remnants (and not "relicts") of former incising valleys that were filled in-situ with sediments while uplifted. This will be rephrased in the revised version of the manuscript.

**ESurfD**
RC1: L406: Yes, but for a relatively short time, this is one of the key points of Whipple et al, 2017 (JGR-ES) paper.

=> We kind of agree with RC1, as the (short or longer) time for return to an equilibrium profile depends on the response time of the river network to this perturbation. This is already further discussed and illustrated in section 5.2 based on this earlier work (Whipple et al 2017, but also Schwanghart and Scherler 2020) and on our observations.

RC1: L412: As earlier, Forte & Whipple, 2018 is the more appropriate reference here as this paper highlights the complications of base level choice.

=> As mentioned previously, Forte and Whipple 2018 provide an expansion of some earlier ideas and conclusions reached in Whipple et al 2017 JGR ES. But we agree that this 2018 paper could also be cited here.

RC1:L663-665: It would be useful perhaps to consider this in the context of the aforemen- tioned Adams et al, 2020 paper. I.e. they demonstrate that the large magnitude variations in precipitation rate have an important control on the scale of the topography (ksn) and its relation to erosion rates. The Gilbert metrics shouldn't be influenced by this, but you've calculated chi assuming static K and precipitation (as most do), but in the context of the Adams result, I wonder if calculating chi with the modern spatially variable precipitation would alter the chi patterns? My hunch would be no, and I don't necessarily think you need to demonstrate this, but I think it would be good to acknowledge that there are pretty significant precipitation gradients and they have been shown to influence topography and the reflection of erosion rates within topography.

=> We thank RC1 for mentioning the Adams et al 2020 paper, with specific focus on how large precipitation variations impact topography and erosions rates, and taking the Bhutan Himalayas as a field example. We agree that Gilbert metrics, with across-divide contrasts in various morphometric parameters, are not to be affected by precipitation gradients as these metrics are local observations and are therefore expected to reflect **ESurfD**
similar background forcing conditions. When calculating transformed chi coordinates and river profiles, RC1 is right in that there is the underlying assumption of constant precipitation rates over the drainage basin (as drainage area is being taken as a proxy for river discharge) - an assumption not verified here, and in fact in most large-scale drainage basins. Locally higher precipitation rates may mistakenly lead to chi profiles resembling those related to drainage area gain (more water), and vice versa. In the case of our morphometric analysis of the Bhutan Himalayas, we do believe that this classical limitation of transformed coordinates does not, however, impact our results. Indeed, strong north-south precipitation variations are found similarly everywhere in Bhutan (See for instance Figure 1 of supplementary material of Grujic et al 2006 that clearly illustrates this). As large-scale rivers in Bhutan are flowing north-south, perpendicular to this climatic trend, they are all similarly affected: the cross-comparison of river profiles, as done in our study, is therefore permitted. In the case of secondary tributary streams, these are compared to their trunk stream locally at their confluence, and therefore most often encompassing similar local climatic conditions. Finally, extreme precipitation rates are found in Bhutan only along the mountain front (up to 50 km from the topographic front T1), is south of the region of greatest interest of our study of the morphology of the mountain hinterland. Rather than demonstrating this and not to lengthen the paper with unnecessary calculations, we will add some clarification to this in section 5.1 on the potential limits of our approach.

RC1:L745-766: A fundamental problem with applying the Yang et al hypothesis and/or the Willett criteria for recognizing area loss/gain in chi-transformed river profiles to this landscape is the hypothesized presence of relatively discrete structural breaks (i.e. the blind duplex of Adams). This fundamentally violates some of the underlying assumptions in a pretty big way. More specifically, in chi-transformed space, a river profile responding to a growing duplex is going to look like a river having gained area. The key as you allude to elsewhere is the spatial consistency of the pattern, and thus probably not all of the area gain signatures are tectonic related, but some might be. You ultimately exercise caution in terms of applying the area loss feedback mechanism,

**ESurfD**
which is warranted, but I think a more nuanced look at what you might expect in a structurally complex setting like this is important.

=> We agree with RC1 in that linear transformed river profiles (as those illustrated in Figure 4) are expected in the case of constant forcing and boundary conditions (uplift, climate, lithology...) throughout the river course. In the case of locally higher uplift, as expected over a blind ramp, the river steepness (and therefore the river slope in chi coordinates) is locally higher, so that the river profile moves "higher" in transformed chi plots. This was already stated and explained in our section 3 (Initial text lines 385-386: "For U and K varying along the profile of the river, steeper (gentler) segments in chi profiles relate either to locally higher (lower) uplift rate or to lower (higher) erodibility."). Such pattern could indeed be mistakenly taken as reflecting river captures. However, to avoid this confusion in the analysis of chi profiles, it is important to define a reference equilibrium profile, and only river profiles that move above this equilibrium reference, whatever the slope and the pattern of this reference, should be considered as reflecting drainage area gain by captures. This is why we do not conclude that there are river captures only from the high steep chi profiles of some of the rivers, but by comparing these profiles to a reference local profile. This reference profile is either that of the main trunk stream when analyzing the profiles of secondary tributary streams (ex: Figure 9, section 4.3), or that or large Himalayan rivers such as the Puna Tsang or the Kuri Chhu over the same region when analyzing the profiles of the Wang and Chamkhar Chhu (ex: Figure 5, sections 4.1 and 4.2). In the case of uplift over a blind ramp or in the case of any other structural complexity as found in tectonically active areas-, all profiles should be affected, and only conclusions relying on this above-mentioned cross-comparison are too be considered. This can be further and additionnally clarified in the methodology section (section 3.2.4 and Figure 4) to avoid any misunderstanding.

RC1:L865-867: This is confusing, as Adams et al, 2016 explicitly argues for the knickpoints generated by the duplex to be fixed in longitudinal position (e.g. their figure 6 or figure 10), which seems consistent with your observations, but you cast it as though

**ESurfD**
this is an observation that disagrees with the Adams et al, 2016 model?

=> See previous answer above. Even after re-reading carefully Adams et al 2016, we do not agree with RC1. In this manuscript knickpoints are mentioned throughout the manuscript as migrating upstream, and this is further illustrated in the figures mentioned by RC1 as detailed previously.

RC1:L868: See prior comments about the confusing use of relict-landscapes.

=> See prior responses to these comments.

RC1:L885-887: As noted previously, this seems to at least misrepresent the conclusions of some of the prior work.

=> As mentioned and proposed above, this will be easily rephrased to make better credit to previous work.

**ESurfD**

---

## Author Comment (AC2) · 4 Mar 2021

NB: Hereafter, comments posted by Wolfgang Schwanghart (RC2) are preceded by "RC2", and are followed by the authors' response (preceded by =>).

RC2: I enjoyed reading the manuscript by Simoes et al. It summarizes the controversies around the enigmatic high-elevation low-relief landscapes in Bhutan. Based on geomorphometric analysis of river profiles and drainage divides, the authors emphasize the role of divide migration in shaping the low-relief regions and conclude that existing denudation rates should be reevaluated given these dynamics. Overall, the manuscript is well written, although lengthy at times. Figures have a high quality, but

could be simplified and better annotated for better readability (see comment below). The number of figures seems adequate, but some of the plots appear in very similar form twice (for example Fig. 7a and the map in Fig. 8). This could be avoided. The methods are sound and described in a way that they are reproducible. In parts, the results are intermingled with interpretations which would be better placed in the discussion (e.g. 502-506).

=> We thank RC2 for his positive appreciation of our work and of our analyses, and thank him for providing interesting comments and suggestions that will help improve the manuscript. We agree that some of the figures could be better annotated for an easier reading, and that some may be merged together (ex: Figures 7 and 8 could be merged into 1 single figure, the same for Figures 10 and 11). In our subsequent careful revision of the manuscript, we will make our best to separate results and their interpretations, from discussion whenether appropriate.

RC2: As reviewer #1 notes, I also find it difficult to see how the results of this study cor- roborate or contradict the findings of the studies by Adams et al. Moreover, I find it difficult to follow why other concepts of tectonic rejuvenation (Duncan et al. 2003) are dismissed, based on the grounds that there is an absence of a coeherent wave of incision. Shouldn't it be expected that such a coherent wave is missing given that drainage divide mobility may be a process that prevails throughout this landscape?

=> In the case of how our results compare to those of Adams et al 2016, we suggest to see our detailed response to the various comments by reviewer #1 (AC1). Our morphometric analyses get to the conclusion that the peculiar morphologies in Bhutan are a response to active uplift in the mountain hinterland - a conclusion already reached by Adams et al (2016) from a different perspective. When compared to this earlier work, we additionally document the dynamics of this response, with river captures and migrating divide (ie instability of the river network), an observation that was not reached by previous authors and that allows for refining their initial ideas. The comparison to this previous study will be rephrased to make clear credit to their initial findings, and to

**ESurfD**
clarify our input and step forward.

The idea that an upstream coherent wave of incision may not be straightforward to observe and extract in the case of divide mobility throughout the landscape is quite interesting, and should be considered indeed - we do thank RC2 for this interesting comment! We agree that pervasive area gain/loss by divide migration may alter transformed river profiles in such a way that it may be difficult to observe a potential wave of incision migrating upstream, as expected theoretically (Figure 4b). This has been somehow illustrated by Schwanghart and Scherler 2020 in the case of the Parachute Creek Basin (Co, USA), where the dispersion in the knickpoints related to the upstream migration of a wave of incision is interpreted to relate to coeval progressive changes in upstream drainage area related to divide migration. However, in the case of the Bhutan Himalayas where erodibility is much greater, the landscape appears to respond relatively fast to progressive changes in drainage conditions (section 5.2) so that progressive divide migration is not expected to alter profoundly transformed river profiles. This is not the case for captures and sudden large gains/losses of drainage area, which leave a greater imprint on transformed river profiles (ex: Giachetta and Willett 2018). Given this, we agree that these limits should be further mentioned (section 5.1), and that our conclusions on the absence of a coherent wave of incision from chi profiles should be further nuanced (ex: absence of clear evidence for such an upstream migrating wave of incision - rather than concluding that this wave of incision is absent).

It should be however noticed that transformed river profiles are much more diverse, and that major knickpoints are much more dispersed in chi, altitude AND amplitude, when compared to the Parachute Creek Basin (Schwanghart and Scherler 2000) or the Upper Blue Nile (Giachetta and Willett, 2018) examples, so that a coherent wave of incision migrating upstream into a relict landscape or uplifted terrane remains a weak potential mechanism. Following on this, the earlier interpretation of Duncan et al 2003 considers the large-scale uplift and rejuvenation of the whole mountain range in

Bhutan, and not only locally along the longitudinal band where we observe the morphologic dynamics described in our manuscript. As such the more local tectonic rejuvenation proposed by Adams et al 2016, with recent local uplift over a blind ramp/duplex in the Bhutan hinterland is a much more plausible interpretation. Therefore the absence of evidence for a coherent wave of incision migrating upstream (despite its limits) AND the spatial organization of the geomorphological dynamics documented here are the best arguments to dismiss the earlier interpretation by Duncan et al 2003. This will be better explained in the revised version of the manuscript.

RC2: The observation that catchments downstream of knickpoints are expanding is intriguing, but the mechanism that generates the expansion remains unclear. The studies by Struth et al. (2019) and Giachetta and Willett (2018) are referenced in this context, but these studies show examples where expansion happens downstream of areas with internal drainage and that were integrated in the flow network. Are endorheic basins a possible explanation for the preservation of these landscapes? And if not (which is quite likely given the humid climate), what could be an alternative intepretation? An hypothesis that might be brought forward could be the availability of sediments mobilized from the alluvial plains upstream that would act as tools accelerating incision downstream which would propagate towards the divides.

=> We agree that the studies we refer to (Struth et al 2019 in particular here, but also Giachetta and Willett 2018) both report captures of internal drainages. As mentioned in our manuscript (lines 750-752), because there is no clear evidence of drainage area loss in transformed river profiles, even in the case of potential large-scale captures such as for the Wang or Chamkhar Chhu, we propose that these captures may have been at the expense of the low-relief regions themselves if once isolated from the main river network - ie supposing that they may have been temporary internally draining hanging valleys, before capture. After this comment and to further document such potential large-scale captures, we have been exploring this idea following the above-cited studies, by calculating the theoretical transformed profiles of the possible proto-

Wang and -Chamkhar rivers in the case that the drainage area upstream of their major knickpoint has been captured. Such transformed profiles are broadly concordant to those of the large rivers that have supposedly equilibrated (ie Kuri, Puna Tsang) - further supporting our interpretation of large-scale captures. This will be added in the revision of the manuscript.

Such large-scale captures, possibly of internal basins, may be surprising in the case of a tropical climate, even though some internal drainages exist or may have existed in similar climatic and tectonic contexts, for instance in the region of the Sun Moon Lake reentrant of Central Taiwan (ex: Toushe Basin, or formerly Yuchi basin south of the Sun Moon Lake), with suspected similar captures of parts of this region in the past (Simoes et al 2014).

We agree with RC2 that additional drainage area by capture may enhance incision and base level lowering by adding discharge but also by remobilizing sediments (and therefore tools) from the captured upstream alluvial plains. This would certainly favor the incision of tributaries downstream of major knickpoints, and therefore the outward expansion of the downstream drainage area by divide migration. The tool effects of sediments drained out of the captured alluvial plains do however not leave a clear imprint on our transformed profiles (Figure S8, and lines 678-684) as expected after the work of Giachetta and Willett 2018, so this mechanism may not be dominant here. Additional comments on this will be added in the revised manuscript.

RC2: I find it difficult to read some of the figures. The combination of a grayscale depiction of topographic relief (which is quite printer-unfriendly), and colored networks makes some maps really busy and difficult to read. The colored stream networks (e.g. in Fig. 2c and 5) have variations in blue and green that are quite subtle or not resolved by my printer. Consider to label the river profiles in the plots rather then using a legend.

=> Topographic relief is commonly and classically depicted with gray-scale for an easier reading of other metrics (with colors) over relief maps. Therefore we'd rather keep

this gray-scaling to keep it simple. In the case of the colored networks on maps and of colored river profiles, an easier reading will be hopefully permitted by additionally labeling rivers on maps or on profiles, and/or by thickening some of the lines in profiles in the subsequent revised version of the figures

RC2: In addition to above major comments, I have numerous minor comments listed below:

=> Most of the subsequent minor comments are suggestions of rephrasing. Unless mentioned and justified, these will be easily implemented in the subsequent revision of the manuscript. We do thank RC2 for his suggestions and improvements.

RC2: 29: Remove "indeed". In general, the text contains numerous filler words, which could be avoided.

RC2: 35: Remove "first-order". I have seen this term a couple of times in this manuscript, but I don't know what it actually means in most contexts. For example, in line 63, I don't understand the term "first-order consistency".

=> "First-order" refers to the fact that there is a general broad consistency in the patterns, although some variations in the details. We will simplify whenever appropriate.

RC2: 253: the term "rather relative" is quite vague, as is the term "rather similar" in line 256.

RC2:394: remove 'long-distance'

=> We do agree, the term 'longitudinal profile' is here more appropriate, and more consistent with the terminology used throughout the text.

RC2:395: migrate upstream in response

RC2:395: what do you mean by 'common process'.

=> We mean a common mechanism, here a common change in forcing or boundary

conditions. This will be rephrased and clarified.

RC2:396: perhaps rephrase "are expected to cluster in transformed coordinates".

RC2:411: , however,

RC2:419: Consider shortening this sentence: These complementary methods enable a more careful assessment of divide migration direction and drainage network reorganization.

RC2:424: Perhaps rephrase: Based on visual interpretation of longitudinal and chi profiles, we identify three profile types of major rivers in Bhutan.

RC2:425: Avoid the term 'simple'. Rather write that these profiles are concave upward with no remarkable knickpoint.

RC2:426: Remove 'rivers like'

RC2:433: 'intermediate characteristics' is a bit vague.

RC2:441: above 3800 m

RC2:456: Not sure what "better organized" means

=> We mean here that the various trends in river profiles are more visible and differentiated in transformed coordinates when compared to longitudinal profiles. This will be rephrased.

RC2: 459: Remove 'clearly' twice

RC2:462: Remove 'first-order'

RC2:465: You may better write "analyze the geometry of". The dynamics will be inferred from the geometry.

RC2:476: The sentence is vague: rivers compare well and are colinear to the very first-order. I am also not sure what you mean by 'first order' as used in the next sentence.

In addition, this part mixes observations (or results) and interpretation.

=> "First-order" is here used as "broadly", ie profiles are broadly concordant despite some secondary variations when getting into details. As of mixing observations and interpretations, this is quite minor as only sentence "Altogether, these observations suggest that these rivers share and have adjusted to the first order to similar tectonic and/or climatic forcing conditions, even though located throughout Bhutan" is an interpretation here, an interpretation that needs to be associated with results and not with discussion. So we'd rather keep it here or eventually move it to section 4.5 (Results: general conclusions) where we summarize our main findings and interpretations before getting into discussion, depending on whether repetition of our various observations and arguments could be avoided by doing so.

RC2: 479: On which basis do you judge that a knickpoint chi-value is discordant from an- other. Consider providing quantitative evidence. One possible way to report these differences in chi values could involve calculating the necessary change in area required so that the locations of knickpoints are the same in chi space. This would allow readers to appreciate the differences in knickpoint locations and would provide a way to eventually exclude or consider divide dynamics as potential mechanism that creates the differences in knickpoint locations.

=> We agree with the fact that the variability of natural conditions, with respect to theory, may lead to some secondary discordance in the details of chi profiles, even though theoretically concordant. This is illustrated in Figure 4b. Defining an acceptable degree of discordance in profiles is a solution to quantitatively define discordant from concordant profiles, but the definition of such a threshold is by essence totally arbitrary, whether this threshold is defined from the observed average dispersion of chi coordinates of knickpoints (as in Schwanghart and Scherler 2020), or from the definition of an acceptable calculated gain or loss of drainage area needed to have the profiles considered as concordant (as suggested here). As clearly visible in Figure 5b, all river profiles south of T3 are discordant, with variable positions of knickpoints, in terms of chi coordinates (dispersion over 1000 m) AND in terms of amplitudes or altitudes (from altitudes of 1200 m to 2700 m) - a situation quite different from that depicted by Schwanghart and Scherler 2020 for the Parachute Creek Basin (Co, USA), where knickpoints are dispersed over a same range of chi values, but clustered around an altitude of 2400 m. We will better explain this in the revised version of the manuscript. In our case study, the situation is therefore quite straightforward, and we'd rather keep it simple.

Following this comment, rather than calculating the drainage area gain/loss needed to have profiles more concordant, we calculated the theoretical transformed profiles of major rivers (Wang and Chamkhar Chhu) before a possible capture of the area upstream of their major knickpoint, and found that the profiles of these theoretical proto-rivers are clearly more concordant with those of the other large rivers (Kuri, Puna Tsang) over the region south of T3. This further supports our interpretations and we thank RC2 for giving us indirectly this idea. This will be added in the revised version of the manuscript.

RC2: 497: remove words like "clearly"

RC2:502: this paragraph should be better placed in the discussion

RC2:549: Avoid the term "dramatic" (which is found several times in the manuscript).

=> The term 'dramatic' has been used to refer to potential large-scale river captures along major rivers. It may in fact not be appropriate here and a clear reference to the spatial scale ('large-scale' instead of 'dramatic') is probably better.

RC2: 550: While expansion is the right term, I don't like the term contraction is this context, because it implies that there are processes that exert a stringent force. I would rather use "specific pattern of drainage area loss and expansion".

RC2:565: Better place this sentence in the discussion.

RC2:640: Such summaries are generally helpful. However, you may consider moving it to the beginning of the discussion, also.

**ESurfD**

Interactive
comment

=> We'd rather keep this summary in the results section, as it provides the basic interpretations that can be driven directly from our various observations. This summary section could in fact take up the text that has been previously suggested to be moved from results to discussion.

These interpretations are rather straightforward from observations, and should be separated from a discussion section devoted to discussing the limits of the approaches/interpretations, but also the implications of our results and interpretations in moving a step forward. In fact, we distinguish results/interpretations from discussion - and do not wish to mix direct interpretations with discussion. A solution could be also to modify the title of this section to "Summary of key results and their interpretations", or something alike.

RC2: 645: robustly? Robust in statistics usually means insensitive to outliers. I am not sure what it means here.

=> 'Robustly' is used in the sense that the comparison between trunk and tributary profiles is more rigorous than between various trunk channels that may not share the same outlet - and therefore interpretations less weak. 'Rigorously' may be a more appropriate alternative.

RC2: 659: remove "whatever the dimensions of their drainage basins"

RC2:691: replace "the classical" with "known"

RC2:692: replace "more generally speaking" with "in general"

RC2:699: rephrase to avoid "dramatic"

RC2:756: what are "stable soils"? There is no Fig. 3g.

=> We meant here well-developed soils, as expected in places where weathering is dominant over mechanical erosion. There is indeed no figure 3g, and it will be corrected to Figures 3b-d

**ESurfD**
RC2: 835: This assertion of an "absence of a coherent wave" needs better quantitative justifi- cation, as mentioned above. And given that divide dynamics are an important process, isn't that what you would expect irrespective of the absence or presence of a large- scale tectonic or climate signal?

=> As explained previously, the definition of a threshold to distinguish acceptable con- cordant profiles (within dispersion) from discordant profiles is arbitrary and we'd rather keep things more simple, in particular given the large and obvious dispersion in chi, amplitudes and altitudes of the knickpoints considered in our study case (see Figures 5b or 9c for instance).

As also answered earlier, we will nuance our conclusions on the absence of a coherent wave of incision (ie absence of evidence for a coherent wave of incision, with respect to what is theoretically expected) as we do agree that the captures observed throughout the studied landscape may weaken our related previous interpretations to some extent.

---

## Author Response (AR1)

Martine Simoes
Institut de physique du Globe de Paris - Université de Paris
1 rue Jussieu
75238 Paris cedex 05
France
e-mail: simoes@ipgp.fr
tel: (+33) (0)1 83 95 76 26

Earth Surface Dynamics

Paris, April 8th 2021

Dear Editor and Associate Editor,

Please find enclosed the revised manuscript "*Topographic Disequilibrium, landscape dynamics and active tectonics: an example from the Bhutan Himalayas*" by Martine Simoes, Timothée Sassolas-Serrayet, Rodolphe Cattin, Romain Le Roux-Mallouf, Matthieu Ferry and Dowchu Drukpa, submitted to *Earth Surface Dynamics*.

We received two reviews, by an anonymous reviewer (RC1) and by Wolfgang Schwanghart (RC2), which helped improve and clarify the presentation of our work. We answered all their comments and posted our answers in the discussion appended to our manuscript (AC1 and AC2, respectively). Hereafter, we recall all their comments, recall and complement our answers, and indicate the subsequent associated revisions of our manuscript.

We hope that you'll find now our manuscript suitable for publication in *Earth Surface Dynamics*.

Sincerely,

Martine Simoes
(on behalf of all co-authors)
* * *
**Comments by Anonymous Reviewer #1
and associated answers/corrections**

Hereafter, all comments posted by Anonymous Referee #1 (RC1) are indicated in italic and preceded by "RC1", and are followed by the authors' response (preceded by =>).

*RC1: In the submitted paper "Topographic disequilibrium, landscape dynamics, and active tectonics: an example from the Bhutan Himalayas", Simoes et al perform a topographic analysis of the Bhutanese Himalaya, with a special focus on the low-relief, high-elevation surfaces that have attracted prior attention in this portion of the range. They primarily approach this landscape from the perspective of evaluating drainage divide instability and the extent to which this either complicates prior results or helps to demonstrate what may be driving the landscape form in this region.*
*Technically the paper is fine, the analyses are appropriate and the individual interpretations*

*that flow from these analyses are mostly warranted and/or logical.*

=> We do sincerely thank RC1 for appreciating the quality of our analyses and of our work on the dynamics of the river network in the hinterland of the Bhutan Himalaya.

*RC1: My main issue is that the motivation of the paper, and their addressing of this motivation in the discussion/conclusion, seems a bit problematic. They set up the paper by describing the landscape, its interesting morphology, and some of the prior tectonic and geomorphic interpretations. The problem is that they end up misrepresenting the interpretation from Adams et al, 2016 in the motivation and then after their analysis essentially confirm most of what Adams was arguing for, but still indicating that it wasn't what Adams was arguing for (e.g. they suggest that the Adams model was inconsistent with nearly static knickpoints, where in the Adams model explicitly argued for nearly static knickpoints). Similarly, they describe the Adams paper as arguing for the preservation of a "relict- landscape", where in detail the Adams paper explicitly argues against a relict landscape preservation hypothesis (some of this may be semantic, i.e. it seems like they are using an odd, non-standard definition of relict landscapes which differs from what is normally used, so this could be fixed by clarifying what they mean by specific terms).*

=> We do regret the misunderstanding on how we decribe how Adams et al (2016) meet our final conclusions on the fact that active uplift is the most probable supporting mechanism for the observed peculiar morphologies, even though in the details some of their model inferences deviate from our observations. The work by Adams et al (2016) appears as the most advanced interpretation of these peculiar morphologies and we wish to give good credit to their work. By emphasizing the differences between their model and our observations, we also wish to point the way to move forward in the future in the modeling of the observed morphologies - and not to discredit their work.

We have accordingly substantially modified section 5.3.4 where we discuss previous interpretations in ligh of our findings. More specifically, the title has been modified to "Discussing previous interpretations in ligh of our findings" (line 848), instead of "Testing previous interpretations", which may sound like setting a controversy. Also, we took good care to indicate that Adams et al (2016) reached the same idea that uplift in the mountain hinterland was needed to support the observed morphologies, interestingly coming at it from a different angle (lines 873-878). We also indicated clearly the details of their model and inferences that do not fit our observations and deductions, and use this to propose a way to move forward (lines 879-894). These differences arise most probably because these authors set their conclusions by comparing the observed morphologies to a landscape evolution model where the geometry of the network is fixed. Future models will need to integrate the mobility and dynamics of the network as a possible landscape response.

Additionally, in the details:

- static versus migrating knickpoints: we disagree with RC1. Indeed, Adams et al (2016) suggest that major knickpoints migrate upstream. This is repeatedly stated in their manuscript that we have carefully re-read.

Some examples citing the text by Adams et al (2016), just by a simple search for the words "migrating knickpoint" throughout the text (pages refer to the PDF): "*Failure to match the rising local base- level set by the migrating knickpoints with a similar deposition rate would have led to a defeated, ponded river and an internally drained basin* " (p15); " *The stippled pattern marks the packages of sediment accumulating upstream of a migrating convex*

*knickpoint (black dot) and forming the migrating concave knickpoint upstream (white dot) "* (caption of Figure 6); " *Comparisons with our landscape evolution model and the observed sediment deposits both suggest that the low-relief landscapes of Bhutan were actively aggrading as they adjusted to the local baselevel rise created by a migrating convex knickpoint* " and " *I is the incision rate into bedrock at the position of the migrating convex knickpoint*"(p17); " *Our landscape evolution experiment also supports the hypothesis that such low-relief landscapes are transient features whose positions are controlled by head- ward migrating, convex knickpoints, as evident from the dichotomy in erosion rates between the low-relief landscapes and adjacent canyons.* " (p. 23).

This is also illustrated in figures 6 and 11 of Adams et al (2016). Indeed in their figure 6, the knickpoint migrates from position ~90 km in b) to position ~110 km in c) while the model evolves in time; in their figure 11, the knickpoint migrates upstream from positions ~12 km (river 2) and ~15 km (river 1) to positions ~15 km (river 2) and ~18 km (river 1), respectively. This migration remains limited - even though the time that separates each of these model snapshots is not reported - , but goes together with the idea stated in the manuscript that knickpoints migrate ustream in the model.

- relict landscapes: we agree with RC1 that our initial terminology and phrasing may have been confusing and misinterpreted. By "relict landscapes", we referred here to the fact that former valleys of the mountain hinterland had been preserved (even though subsequently filled in-situ with sediments) and uplifted in Adams' model. This lead Adams et al (2016) to use the uplifted position of these alluvial valleys as a marker of uplift above a theoretical initial river profile. In fact, in their model, the overall shape of the valleys are remnants of former incisional valleys (explaining that we used the term "relict" for 'remnant', initially), and that only alluvial filling occurred in-situ during uplift (what Adams et al 2016 termed 'in situ formation of the valleys' - probably also confusing). We recognize that the term "relict landscape", classically used in the case of landscapes formed along mountain foothills and grading to the foreland, is not adapted here.

We corrected for this in the text when referring to Adams et al (2016) work (lines 269-270, 885-88). Also to avoid any confusion with the term 'relict landscape', we substantially modified section 5.3.2 on the characteristics and possible interpretations of the low-relief regions. We believe that our point is now much clearer.

*RC1:* *There is certainly value in documenting some of the interesting and nuanced drainage network reorganizations that are occurring in this landscape, but the paper suffers from seeming to set up sort of a false controversy (and it is unfair to the Adams paper in that ultimately, most of the observations here confirm, or are consistent with, hypotheses put forward in the Adams paper). I think recasting the introduction / conclusion of the paper to be less about testing or addressing a controversy and more about exploring another interesting aspect of this landscape that wasn't really addressed in the prior work by Adams et al (various years), i.e. drainage network instability, and thinking about how this is being driven / influenced by the tectonic context seems much more appropriate. Ultimately, coming at it from this approach may allow for more interesting and meaningful interpretations and/or implications.*

=> We regret that our work has been seen as setting any kind of controversy, as our objectives were not those. Indeed, we aimed at documenting and understanding the dynamics of the river network and the relative time scales for landscape response from the particular example of the Bhutan Himalaya where out-of-equilibrium morphologies have been documented. Our study also provides an interesting field example where the classical use of morphology to derive rates

of active tectonics is to be done with great caution.

When comparing our results to previous work and interpretations, we wish to give good credit to all previous work, and in particular to the work by Adams et al (2016) that proposed up to now the best model that fits most of our observations. Even though we agree with Adams et al 2016 on the idea of active uplift in the mountain hinterland, our results emphasize the limits of their model, and by doing so aims at pointing out future directions of work, in particular by proposing to better include the dynamics of the drainage network when modeling the lansdcape response to active tectonics.

We have modified the end of our introduction to clarify our objectives (lines 113-123), and, as indicated previously, the way we discuss previous work by Adams et al (2016) (lines 873-900).

*RC1: L70-71: This statement at least does not reflect one of your cited references, i.e. Adams et al, 2016 argue for in-situ development of the low-relief surfaces from blind duplexing, which they argue may be structurally linked to the development of the Shillong plateau, but definitely is not representative of "relicts of former climatic or tectonic conditions".*

=> As stated above, we agree that using the word "relict" in the case of the Adams' model may be confusing and should be avoided. We have corrected for this (lines 66-68).

*RC1: L118: The Gilbert metrics are formally defined in Forte & Whipple, 2018, not in the Whipple et al, 2017 JGR-ES paper.*

=> We do not fully agree with RC1. The idea of the Gilbert metrics was first proposed in Whipple et al 2017 JGR Earth Surface (see for instance section 5 of this 2017 manuscript "Topographic Metrics for Recognizing Mobile Divide", p 263-265 ; in addition to their section 7.2. "Utility of Topographic Metrics of Erosion and Divide Mobility", p 269-270) - even though these metrics were not initially termed "Gilbert metrics". These metrics were named as such, further expanded and discussed in the Forte and Whipple 2018 paper. We added this citation for Forte and Whipple (2018) wherever missing.

*RC1: L145-144: You might also consider citing the recent Adams et al, 2020 (Adams, B.A., Whipple, K.X., Forte, A.M., Heimsath, A.M., Hodges, K.V., 2020. Cli- mate controls on erosion in tectonically active landscapes. Science Advances 6. https://doi.org/10.1126/sciadv.aaz3166) as their analysis of this region is also consistent with a relative invariant erodibility for much of the Bhutan region.*

=> We thank RC1 for this suggestion, which has been integrated in our revision (line 144).

*RC1:L281-282: "low-relief hanging fill valleys can be interpreted as relict landscapes formed locally", this seems like a very odd way to describe a landscape, that in the interpretation you're describing, is actively maintained by uplift of blind duplexes and the original authors describe as forming in-situ and explicitly reject the idea of these being "relict landscapes" in the traditional sense. I would consider rewording this to avoid confusion.*

=> As stated and explained in detail above, we recognize that the term "relict" was confusing

and not used following the classical meaning. This has been rephrased (lines 269-270).

*RC1:L293-294: See previous point, i.e. at least when considering the Adams model, they explicitly reject the idea of these being relict landscapes, at least in the way this term is typically used (e.g. the Whipple et al – Willett et al paper/comment and reply chain that you cite). I think you either need to reword this and other places or be much more explicit about how you are using/defining relict landscape, because this seems to be a non-standard way of describing them and it is (1) confusing and (2) misrepresents the results of previous work if you apply the more standard definition of relict landscapes.*

=> See previous answers above. Indeed, we agree that "relict landscape" was not meant in our manuscript in the classical way, but rather in the sense that alluvial valleys are interpreted as remnants (and not "relicts") of former incising valleys that were filled in-situ with sediments while uplifted. This has been corrected (lines 280-283), and section 5.3.2 has been modified to avoid further confusion.

*RC1: L406: Yes, but for a relatively short time, this is one of the key points of Whipple et al, 2017 (JGR-ES) paper.*

=> We kind of agree with RC1, as the (short or longer) time for return to an equilibrium profile depends on the response time of the river network to this perturbation. This is already further discussed and illustrated in section 5.2 based on this earlier work (Whipple et al 2017, but also Schwanghart and Scherler 2020) and on our observations. We modified to clarify that these are transient features (lines 395-396).

*RC1: L412: As earlier, Forte & Whipple, 2018 is the more appropriate reference here as this paper highlights the complications of base level choice.*

=> As mentioned previously, Forte and Whipple 2018 provide an expansion of some earlier ideas and conclusions reached in Whipple et al 2017 JGR ES. But we agree that this 2018 paper could also be cited here. See previous corrections.

*RC1:L663-665: It would be useful perhaps to consider this in the context of the aforementioned Adams et al, 2020 paper. I.e. they demonstrate that the large magnitude variations in precipitation rate have an important control on the scale of the topography (ksn) and its relation to erosion rates. The Gilbert metrics shouldn't be influenced by this, but you've calculated chi assuming static K and precipitation (as most do), but in the context of the Adams result, I wonder if calculating chi with the modern spatially variable precipitation would alter the chi patterns? My hunch would be no, and I don't necessarily think you need to demonstrate this, but I think it would be good to acknowledge that there are pretty significant precipitation gradients and they have been shown to influence topography and the reflection of erosion rates within topography.*

=> We thank RC1 for mentioning the Adams et al 2020 paper, with specific focus on how large precipitation variations impact topography and erosions rates, and taking the Bhutan Himalayas

as a field example.

We agree that Gilbert metrics, with across-divide contrasts in various morphometric parameters, are not to be affected by precipitation gradients as these metrics are local observations and are therefore expected to reflect similar background forcing conditions. When calculating transformed chi coordinates and river profiles, RC1 is right in that there is the underlying assumption of constant precipitation rates over the drainage basin (as drainage area is being taken as a proxy for river discharge) - an assumption not verified here, and in fact in most large-scale drainage basins. Locally higher precipitation rates may mistakenly lead to chi profiles resembling those related to drainage area gain (more water), and vice versa. In the case of our morphometric analysis of the Bhutan Himalayas, we do believe that this classical limitation of transformed coordinates does not, however, impact our results. Indeed, strong north-south precipitation variations are found similarly everywhere in Bhutan (See for instance Figure 1 of supplementary material of Grujic et al 2006 that clearly illustrates this). As large-scale rivers in Bhutan are flowing north-south, perpendicular to this climatic trend, they are all similarly affected: the cross-comparison of river profiles, as done in our study, is therefore permitted. In the case of secondary tributary streams, these are compared to their trunk stream locally at their confluence, and therefore most often encompassing similar local climatic conditions. Finally, extreme precipitation rates are found in Bhutan only along the mountain front (up to 50 km from the topographic front T1), ie south of the region of greatest interest of our study of the morphology of the mountain hinterland.

Rather than demonstrating this and not to lengthen the paper with unnecessary calculations, we added some clarifications to this in section 5.1.2 on the potential limits of our approach (lines 698-707).

*RC1:L745-766: A fundamental problem with applying the Yang et al hypothesis and/or the Willett criteria for recognizing area loss/gain in chi-transformed river profiles to this landscape is the hypothesized presence of relatively discrete structural breaks (i.e. the blind duplex of Adams). This fundamentally violates some of the underlying assumptions in a pretty big way. More specifically, in chi-transformed space, a river profile responding to a growing duplex is going to look like a river having gained area. The key as you allude to elsewhere is the spatial consistency of the pattern, and thus probably not all of the area gain signatures are tectonic related, but some might be. You ultimately exercise caution in terms of applying the area loss feedback mechanism, which is warranted, but I think a more nuanced look at what you might expect in a structurally complex setting like this is important.*

=> We agree with RC1 in that linear transformed river profiles (as those illustrated in Figure 4) are expected in the case of constant forcing and boundary conditions (uplift, climate, lithology...) throughout the river course. In the case of locally higher uplift, as expected over a blind ramp, the river steepness (and therefore the river slope in chi coordinates) is locally higher, so that the river profile moves "higher" in transformed chi plots. This was already stated and explained in our section 3 (Lines 374-375). Such pattern could indeed be mistakenly taken as reflecting river captures. However, to avoid this confusion in the analysis of chi profiles, it is important to define a reference equilibrium profile, and only river profiles that move above this equilibrium reference, whatever the slope and the pattern of this reference, should be considered as reflecting drainage area gain by captures. This is why we do not conclude that there are river captures only from the high steep chi profiles of some of the rivers, but by comparing these profiles to a reference local profile (now clarified lines 398-399). This reference profile is either that of the main trunk stream when analyzing the profiles of secondary tributary streams (ex: Figure 8, section 4.3), or that or large Himalayan rivers such as the Puna

Tsang or the Kuri Chhu over the same region when analyzing the profiles of the Wang and Chamkhar Chhu (ex: Figure 5, sections 4.1 and 4.2). In the case of uplift over a blind ramp - or in the case of any other structural complexity as found in tectonically active areas-, all profiles should be affected, and only conclusions relying on this above-mentioned cross-comparison are too be considered.

This is now also further clarified when discussing the previous interpretation by Baillie and Norbu (2004) (lines 856-872 in section 5.3.4), where we point out how lateral (random) variations in uplift can be discriminated from river captures.

*RC1:L865-867: This is confusing, as Adams et al, 2016 explicitly argues for the knickpoints generated by the duplex to be fixed in longitudinal position (e.g. their figure 6 or figure 10), which seems consistent with your observations, but you cast it as though this is an observation that disagrees with the Adams et al, 2016 model?*

=> See previous answer above. Even after re-reading carefully Adams et al 2016, we do not agree with RC1. In this earlier work, knickpoints are mentioned throughout the manuscript as migrating upstream, and this is further illustrated in the figures mentioned by RC1 as detailed previously.

*RC1:L868: See prior comments about the confusing use of relict-landscapes.*

=> See prior answer to these comments on relict landscapes.

*RC1:L885-887: As noted previously, this seems to at least misrepresent the conclusions of some of the prior work.*

=> As mentioned previously, this has been corrected and rephrased.
* * *
**Comments by Wolfgang Schwanghart (Reviewer #2)
and associated answers/corrections**

Hereafter, comments posted by Wolfgang Schwanghart (RC2) are reported in *italic* and preceded by "RC2", and are followed by the authors' response (preceded by =>).

*RC2: I enjoyed reading the manuscript by Simoes et al. It summarizes the controversies around the enigmatic high-elevation low-relief landscapes in Bhutan. Based on geomorphometric analysis of river profiles and drainage divides, the authors emphasize the role of divide migration in shaping the low-relief regions and conclude that existing denudation rates should be reevaluated given these dynamics.*
*Overall, the manuscript is well written, although lengthy at times. Figures have a high quality, but could be simplified and better annotated for better readability (see comment below). The number of figures seems adequate, but some of the plots appear in very similar form twice (for example Fig. 7a and the map in Fig. 8). This could be avoided. The methods are sound and described in a way that they are reproducible. In parts, the results are intermingled with interpretations which would be better placed in the discussion (e.g. 502-506).*

=> We thank RC2 for his positive appreciation of our work and of our analyses, and thank him for providing interesting comments and suggestions that helped improve the manuscript.

We agree that some of the figures may be better annotated for an easier reading (see answer hereafter, in the case of a specific comment on this). Other figures were similar and appeared repetitive. This was the case for former Figures 7 and 8 - which are now reported as Figure 7 in main text and Figure S3 in supplementary material - and for former Figures 10 and 11 - which are now reported as Figure 9 in main text and Figure S6 in supplementary material.

In our careful revision of the manuscript, we attempted to clarify and separate results, interpretations and points of discussion. The various suggestions of interpretations to be moved to a "discussion" section, were however kept mostly in the "results" section to avoid repetitions that would inevitably lengthen the manuscript. In fact, these were direct and straightforward interpretations from results, whereas our 'discusion' section is devoted to discuss the limits of our approach and the implications of our results/interpretations to move a step forward. However, in order to simplify this, we re-wrote the various sections of our results, so as to first simply describe observations, and end each "results" 'section with the straightforward interpretations. This is specified line 466, done in sections 4.2, 4.3 and 4.4, and summarized in section 4.5. We hope this is now clearer.

*RC2: As reviewer #1 notes, I also find it difficult to see how the results of this study corroborate or contradict the findings of the studies by Adams et al. Moreover, I find it difficult to follow why other concepts of tectonic rejuvenation (Duncan et al. 2003) are dismissed, based on the grounds that there is an absence of a coeherent wave of incision. Shouldn't it be expected that such a coherent wave is missing given that drainage divide mobility may be a process that prevails throughout this landscape?*

=> In the case of how our results compare to those of Adams et al (2016), we suggest to see our detailed response and corrections to the various comments by reviewer #1. Our morphometric analyses get to the conclusion that the peculiar morphologies in Bhutan are a response to active uplift in the mountain hinterland - a conclusion already reached by Adams et al (2016) from a different perspective. When compared to this earlier work, we additionally document the dynamics of this response, with river captures and migrating divide (ie instability of the river network), an observation that was not reached by previous authors and that allows for refining their initial ideas. The comparison to this previous study has been rephrased to make clear credit to their initial findings (lines 873-878), and to clarify our input and step forward (lines 879-894).

The idea that an upstream coherent wave of incision may not be straightforward to observe and extract in the case of divide mobility throughout the landscape is quite interesting, and should be considered indeed - we do thank for this interesting comment! We agree that pervasive area gain/loss by divide migration may alter transformed river profiles in such a way that it may be difficult to observe a potential wave of incision migrating upstream, as expected theoretically (Figure 4b). This has been somehow illustrated by Schwanghart and Scherler 2020 in the case of the Parachute Creek Basin (Co, USA), where the dispersion in the knickpoints related to the upstream migration of a wave of incision is interpreted to relate to coeval progressive changes in upstream drainage area related to divide migration, and illustrated theoretically in Giachetta and Willett (2018) in the case of river captures. However, in the case of the Bhutan Himalayas

where erodibility is significant, the landscape appears to respond relatively fast to progressive changes in drainage conditions (section 5.2) so that progressive divide migration is not expected to alter profoundly transformed river profiles - leaving the possibility to extract from the profiles of such streams the signal of a wave of incision if existant. This is not the case for captures and sudden large gains/losses of drainage area, which impact much more transformed river profiles. We added a discussion on these limitations (section 5.1.2, lines 680-690) and rephrased our conclusions on the absence of a coherent wave of incision from chi profiles (ex: absence of clear evidence for such an upstream migrating wave of incision - rather than concluding that this wave of incision is absent) (lines 629-630, 849).

It should be however noticed that transformed river profiles are much more diverse, and that major knickpoints are much more dispersed in chi, altitude AND amplitude, when compared to the Parachute Creek Basin (Schwanghart and Scherler 2000) or the Upper Blue Nile (Giachetta and Willett, 2018) examples, so that a coherent wave of incision migrating upstream into a relict landscape or uplifted terrane remains a weak potential mechanism. Following on this, the earlier interpretation of Duncan et al 2003 considers the large-scale uplift and rejuvenation of the whole mountain range in Bhutan, and not only locally along the longitudinal band where we observe the morphologic dynamics described in our manuscript. As such the more local tectonic rejuvenation proposed by Adams et al 2016, with recent local uplift over a blind ramp/duplex in the Bhutan hinterland is a much more plausible interpretation. Therefore the absence of evidence for a coherent wave of incision migrating upstream (despite its limits) AND the spatial organization of the geomorphological dynamics documented here are the best arguments to dismiss the earlier interpretation by Duncan et al 2003. This is now better explained in the revised version of the manuscript (lines 849-851).

RC2: *The observation that catchments downstream of knickpoints are expanding is intriguing, but the mechanism that generates the expansion remains unclear. The studies by Struth et al. (2019) and Giachetta and Willett (2018) are referenced in this context, but these studies show examples where expansion happens downstream of areas with internal drainage and that were integrated in the flow network. Are endorheic basins a possible explanation for the preservation of these landscapes? And if not (which is quite likely given the humid climate), what could be an alternative intepretation? An hypothesis that might be brought forward could be the availability of sediments mobilized from the alluvial plains upstream that would act as tools accelerating incision downstream which would propagate towards the divides.*

=> We agree that the studies we refer to (Struth et al 2019 in particular here, but also Giachetta and Willett 2018) both report captures of internal drainages. As mentioned in our manuscript (lines 770-772), because there is no clear evidence of drainage area loss in transformed river profiles, even in the case of potential large-scale captures such as for the Wang or Chamkhar Chhu, we propose that these captures may have been at the expense of the low-relief regions themselves if once isolated from the main river network - ie supposing that they may have been temporary internally draining hanging valleys, before capture. After this comment and to further document such potential large-scale captures, we have been exploring this idea following the above-cited studies, by calculating the theoretical transformed profiles of the possible proto-Wang and -Chamkhar rivers in the case that the drainage area upstream of their major knickpoint had been captured (Figure 5c). Such transformed profiles are broadly concordant to those of the large rivers that have supposedly equilibrated (ie Kuri, Puna Tsang) - further supporting our interpretation of large-scale captures (Lines 514-516).

Such large-scale captures, possibly of internal basins, may be surprising in the case of a tropical

climate, even though some internal drainages exist or may have existed in similar climatic and tectonic contexts, for instance in the region of the Sun Moon Lake reentrant of Central Taiwan (ex: Toushe Basin, which is internally drained, or the Yuchi and Puli basins where recent captures are suspected) (see discussion by Simoes et al, 2014). This has been added in our revision (lines 772-775).

We agree that additional drainage area by capture may enhance incision and base level lowering by adding discharge but also by remobilizing sediments (and therefore tools) from the captured upstream alluvial plains. This would certainly favor the incision of tributaries downstream of major knickpoints, and therefore the outward expansion of the downstream drainage area by divide migration. The tool effects of sediments drained out of the captured alluvial plains do however not leave a clear imprint on our transformed profiles (Figure S8, and lines 691-697) as expected after the work of Giachetta and Willett 2018, so this mechanism may not be dominant here, even though we cannot discard it. This discussion has been added lines 743-750.

*RC2: I find it difficult to read some of the figures. The combination of a grayscale depiction of topographic relief (which is quite printer-unfriendly), and colored networks makes some maps really busy and difficult to read. The colored stream networks (e.g. in Fig. 2c and 5) have variations in blue and green that are quite subtle or not resolved by my printer. Consider to label the river profiles in the plots rather then using a legend.*

=> Topographic relief is commonly and classically depicted with gray-scale for an easier reading of other metrics (with colors) over relief maps. Therefore we choose to keep this gray-scaling to keep it simple.
In the case of the colored networks on maps and of colored river profiles, an easier reading will be hopefully permitted by the additional labeling of rivers on maps and on profiles (Figures 1, 2c, 5 and S1).

*RC2: In addition to above major comments, I have numerous minor comments listed below:*

=> Most of the subsequent minor comments are suggestions of rephrasing. Unless mentioned and justified, these were all implemented in the revised manuscript.

*RC2: 29: Remove "indeed". In general, the text contains numerous filler words, which could be avoided.*
*RC2: 35: Remove "first-order". I have seen this term a couple of times in this manuscript, but I don't know what it actually means in most contexts. For example, in line 63, I don't understand the term "first-order consistency".*
*RC2: 253: the term "rather relative" is quite vague, as is the term "rather similar" in line 256.*
*RC2:394: remove 'long-distance'*
*RC2:395: migrate upstream in response*

*RC2:395: what do you mean by 'common process'.*

=> We meant a common mechanism, here a common change in forcing or boundary conditions.

This has been rephrased and clarified (lines 384-385).

*RC2:396: perhaps rephrase "are expected to cluster in transformed coordinates".*
*RC2:411: , however,*
*RC2:419: Consider shortening this sentence: These complementary methods enable a more careful assessment of divide migration direction and drainage network reorganization.*
*RC2:424: Perhaps rephrase: Based on visual interpretation of longitudinal and chi profiles, we identify three profile types of major rivers in Bhutan.*
*RC2:425: Avoid the term 'simple'. Rather write that these profiles are concave upward with no remarkable knickpoint.*
*RC2:426: Remove 'rivers like'*
*RC2:433: 'intermediate characteristics' is a bit vague.*
*RC2:441: above 3800 m*

*RC2:456: Not sure what "better organized" means*

=> We mean here that the various trends in river profiles are more visible and differentiated in transformed coordinates when compared to longitudinal profiles. This has been rephrased (lines 456-457).

*RC2: 459: Remove 'clearly' twice*
*RC2:462: Remove 'first-order'*
*RC2:465: You may better write "analyze the geometry of". The dynamics will be inferred from the geometry.*

*RC2:476: The sentence is vague: rivers compare well and are colinear to the very first- order. I am also not sure what you mean by 'first order' as used in the next sentence. In addition, this part mixes observations (or results) and interpretation.*

=> "First-order" is here (and elsewhere) used as "broadly", ie profiles are broadly concordant despite some secondary variations when getting into details. As of mixing observations and interpretations, we suggest to read our previous answer and corrections to a similar earlier comment.

*RC2: 479: On which basis do you judge that a knickpoint chi-value is discordant from another. Consider providing quantitative evidence. One possible way to report these differences in chi values could involve calculating the necessary change in area required so that the locations of knickpoints are the same in chi space. This would allow readers to appreciate the differences in knickpoint locations and would provide a way to eventually exclude or consider divide dynamics as potential mechanism that creates the differences in knickpoint locations.*

=> We agree with the fact that the variability of natural conditions, with respect to theory, may lead to some secondary discordance in the details of chi profiles, even though theoretically concordant. This is illustrated in Figure 4b. Defining quantitatively an acceptable degree of discordance in profiles is a solution to discriminate discordant from concordant profiles, but the definition of such a threshold is by essence totally arbitrary, whether this threshold is defined from the observed average dispersion of chi coordinates of knickpoints (as in Schwanghart and Scherler 2020), or from the definition of an acceptable calculated gain or loss of drainage area needed to have the profiles considered as concordant (as suggested here). As clearly visible in Figure 5b, all river profiles south of T3 are discordant, with variable positions of knickpoints,

in terms of chi coordinates (dispersion over 1000 m) AND in terms of amplitudes or altitudes (from altitudes of 1200 m to 2700 m) - a situation quite different from that depicted by Schwanghart and Scherler 2020 for the Parachute Creek Basin (Co, USA), where knickpoints are dispersed over a same range of chi values, but clustered around an altitude of 2400 m. In our case study, the situation is therefore quite straightforward, and we'd rather keep it simple. We have justified and further explained this in the revised version of the manuscript (lines 400-404).

Following this comment, rather than calculating the drainage area gain/loss needed to have profiles more concordant, we calculated the theoretical transformed profiles of major rivers (Wang and Chamkhar Chhu) prior to a possible capture of the area upstream of their major knickpoint, and found that the profiles of these theoretical proto-rivers are clearly more concordant with those of the other large rivers (Kuri, Puna Tsang) over the region south of T3. This further supports our interpretations and we thank WS (RC2) for giving us indirectly this idea. This has been added in the revised version of the manuscript (Figure 5c).

*RC2: 497: remove words like "clearly"*
*RC2:502: this paragraph should be better placed in the discussion*

*RC2:549: Avoid the term "dramatic" (which is found several times in the manuscript).*

=> The term 'dramatic' has been used to refer to potential large-scale river captures along major rivers. It may in fact not be appropriate here and a clear reference to the spatial scale ('large-scale' instead of 'dramatic') is probably better. Corrected throughout the text.

*RC2: 550: While expansion is the right term, I don't like the term contraction is this context, because it implies that there are processes that exert a stringent force. I would rather use "specific pattern of drainage area loss and expansion".*
*RC2:565: Better place this sentence in the discussion.*

*RC2:640: Such summaries are generally helpful. However, you may consider moving it to the beginning of the discussion, also.*

=> We'd rather keep this summary in the results section, as it provides the basic straightforward interpretations that can be driven directly from our various observations and results These interpretations should be separated from a discussion section devoted to discussing the limits of the approaches/interpretations, but also the implications of our results and interpretations in moving a step forward. In fact, we distinguish results/interpretations from discussion - and do not wish to mix direct interpretations with discussion. We have modified the title of this section to "Summary of key results " to clarify this.

*RC2: 645: robustly? Robust in statistics usually means insensitive to outliers. I am not sure what it means here.*

=> 'Robustly' is used in the sense that the comparison between trunk and tributary profiles is more rigorous than between various trunk channels that may not share the same outlet - and therefore interpretations less weak. 'Rigorously' is a more appropriate alternative. Corrected

*RC2: 659: remove "whatever the dimensions of their drainage basins"*

*RC2:691: replace "the classical" with "known"*
*RC2:692: replace "more generally speaking" with "in general"*
*RC2:699: rephrase to avoid "dramatic"*

*RC2:756: what are "stable soils"? There is no Fig. 3g.*

=> We meant here well-developed soils, as expected in places where weathering is dominant over mechanical erosion. Corrected

There is indeed no figure 3g, and it has been corrected to Figures 3b-d

*RC2: 835: This assertion of an "absence of a coherent wave" needs better quantitative justification, as mentioned above. And given that divide dynamics are an important process, isn't that what you would expect irrespective of the absence or presence of a large- scale tectonic or climate signal?*

=> See previous answers and corrections.

As explained previously, the definition of a threshold to distinguish acceptable concordant profiles (within dispersion) from discordant profiles is arbitrary and we'd rather keep things simple, in particular given the large and obvious dispersion in chi, amplitudes and altitudes of the knickpoints considered in our study case.

As also answered earlier, we will nuance our conclusions on the absence of a coherent wave of incision (ie absence of evidence for a coherent wave of incision, with respect to what is theoretically expected) as we do agree that the captures observed throughout the studied landscape may weaken to some extent our related previous interpretations.

---

## Author Response (AR2)

Martine Simoes
Institut de physique du Globe de Paris - Université de Paris
1 rue Jussieu
75238 Paris cedex 05
France
e-mail: simoes@ipgp.fr
tel: (+33) (0)1 83 95 76 26

Earth Surface Dynamics

Paris, May 21st 2021

Dear Editor and Associate Editor,

Please find enclosed the manuscript "*Topographic Disequilibrium, landscape dynamics and active tectonics: an example from the Bhutan Himalayas*" by Martine Simoes, Timothée Sassolas-Serrayet, Rodolphe Cattin, Romain Le Roux-Mallouf, Matthieu Ferry and Dowchu Drukpa, submitted to *Earth Surface Dynamics*. It has been slightly revised from the previous corrected version.

We received two reviews, by an anonymous reviewer (RC1) and by Wolfgang Schwanghart (RC2), which helped improve and clarify the presentation of our work. We're pleased that these reviewers appreciated our effort to carefully answer all their previous comments. We thank them for their additional suggestions when reviewing our corrections. We have addressed all these additional comments and provide our answers hereafter.

We hope that you'll find now our manuscript suitable for publication in *Earth Surface Dynamics*.

Sincerely,

Martine Simoes
(on behalf of all co-authors)
* * *
**Comments by Anonymous Reviewer #1
and associated answers/corrections**

Hereafter, all comments posted by Anonymous Referee #1 (RC1) are indicated in italic and preceded by "RC1", and are followed by the authors' response (preceded by =>).

*RC1: I have complete my review of the revised version of "Topographic disequilibrium, landscape dynamics and active tectonics: an example from the Bhutan Himalayas". Generally the manuscript is greatly improved and most of my large scale comments were suitably addressed from my initial review. I have a few minor thoughts for the authors, but nothing that should preclude the acceptance of this after a few tweaks (or maybe without a few tweaks depending on how the authors feel about my thoughts).*

*L324: This is a pretty minor quibble, but I'm not sure how smoothing is significant for the calculation of chi? Ksn, sure, but as long as the smoothing doesn't change the drainage area*

*accumulation (which CRS shouldn't, since it operates exclusively on the extracted stream network), smoothing should have zero impact on the calculation of chi. I think you could just be vague here since you haven't introduced all of the metrics yet and just leave it at saying that smoothing is important for some of the metrics you'll use.*

=> We fully agree with reviewer 1 and thank him for this correction!

*RC1: L793-794: Though for the area-loss mechanism, would you necessarily expect the spatial consistency that you note? You end up essentially using this to favor a tectonic origin, but it seems reasonable to "kill" this idea here, i.e., the extent to which an area loss mechanism is actually consistent.*

=> The area-loss mechanism is a possible response of the river network - and its geomorphic consequence -, not the process driving this response. River captures - and area loss as their counterpart - may occur everywhere in the landscape, but it could be envisioned that these processes may be more prevalent in places where uplift is ongoing, as a geomorphic response to this active uplift. For instance, defeated hanging valleys may end up losing drainage area, locally, as a response to local uplift.
Given this, we do not have the impression that the spatial consistency between all geomorphic features is inconsistent with the area-loss mechanism. We think here that possible geomorphic responses and driving mechanisms should be clearly distinguished in our reasoning. This is clearly stated lines 842-844. No specific correction related to this comment has therefore been made in the manuscript.

*RC1: L895-890: I'm curious how much of an issue you actually expect this to be? You cite this multiple times and imply that it can be quite significant generally and for the Bhutan dataset specifically, but from the cited work, this seems the most problematic for very small basins (e.g., <10 km^2) that are eroding very slowly (with the added caveats of things like erodibility, diffusivity, etc. playing a role). For the data from Adams et al 2016, while some basins are eroding reasonably slow, most of the basins (both the slow and fast eroding ones) are within the size range where this previous analysis suggests that the E/U ratio should be quite close to 1. It's fair to point out that this could be an issue given the documented divide instability you provide here, but it's also important to be honest about how much of an effect you actually expect this to impart, i.e. is the level of distrust you indicate we should have toward this aspect of prior work warranted? At least a qualitative assessment of this should be quite doable since many of the same authors appear on the 2019 paper as this one. This has a little bit of overlap with my comments on the first go around, i.e., that some of the criticism here of prior work seems a little overly harsh. Your study easily stands on its own merits without casting unnecessary aspersions on others work. If you can demonstrate that the majority of basins in previous published datasets may be dramatically influenced by the changes in drainage area and systematically would change the result of what was argued previously, that's one thing, but I think you should either (1) attempt to do this or (2) tone down your criticisms of this aspect.*

=> Once more, our idea here is not to sound negative on this previous work that we sincerely appreciate, but rather to call for caution on the classical interpretation of denudation rates in contexts where the shape of sampled drainage basins is continuously changing by river captures, divide migration and therefore area loss/gain. To this respect we have rephrased lines 926-928. This caution applies to the interpretation by Adams et al 2016 on the timing of uplift from denudation rates, but also on the use of such denudation rates to constrain the

geometry of the underlying MHT, such as in Leroux-Mallouf et al (2015) cited lines 971-973... a paper sharing many co-authors with the one being reviewed and discussed here! We honestly also apply our criticism to ourselves ;-)

A detailed and quantitative appreciation of whether or not these denudation rates are representative or not of uplift rates in Bhutan is out of the scope of this study (our manuscript is already quite long, as emphasized by RC2), and is presently the focus of a specific project in progress. We therefore hope to be able to provide in the near future a more quantitative analysis on this. Such field based analysis would be complementary from our previous modeling analysis. Indeed field is definitely more complex than the simple model in Sassolas-Serrayet et al (2019): natural variability in space and also possibly in time of parameters such as erodibility or climate, horizontal advection in addition to uplift in tectonically active areas such as mountain ranges, etc. Also our previous modeling analysis did not include basins with knickpoints. Because of this the variability and dispersion of denudation rates relative to uplift rates is most probably minimized in our previous numerical work when compared to nature. We also would like to point out that the dispersion and variability of denudation rates relative to uplift rates is also found numerically to be positively correlated with uplift rates, and is therefore expected to be more significant in tectonically uplifting regions (equation 6 of Sassolas-Serrayet et al 2019, see also Figure 9 of Hu et al 2021 in EPSL).

**Comments by Wolfgang Schwanghart (Reviewer #2) and associated answers/corrections**

Hereafter, comments posted by Wolfgang Schwanghart (RC2) are reported in *italic* and preceded by "RC2", and are followed by the authors' response (preceded by =>).

*RC2: First of all, I like to apologize for the delay of my review. In my review of the revised version, I went through the reply to reviewer comments. In addition, I focused on rereading the parts of the text that were changed.*
*I thank the authors for providing careful revisions in response to the comments by the reviewers. I think that Martine Simoes and coauthors have addressed all comments and made changes to the earlier draft of the manuscript where necessary. Overall, these edits have improved the manuscript considerably.*
*With 47 pages, the manuscript is still quite long. This is not to be taken as criticsm, but rather I'd like to encourage the authors to condense the text where possible. Wolfgang Schwanghart Minor comments*

=> Most of the subsequent minor comments are suggestions of rephrasing. Unless mentioned and justified, these were all implemented in the revised manuscript.

*RC2: 10: consider replacing 'most often' with 'commonly'*
*RC2: 11: may however be rare in nature*
*RC2: 13: these drainage dynamics*
*RC2: 13f: 'particular case example' seems like three words meaning the same thing. Why not simply: Here, we document these drainage dynamics in the Bhutan Himalayas, where out-of-equilibrium morphologies have been noticed from major river knickpoints and high-altitude low-relief regions in the mountain hinterland.*

*RC2: 19: these dynamics. Moreover, either landscape response is rapid, or time scale is short. Consider removing time scale*
*RC2: 28: remove "of uplifted terranes".*

*RC2: 320: In order to let others replicate your analysis, please provide tau (the quantile) and K (the smoothness parameter) used in the CRS function.*

=> This is now indicated (lines 340-341)

*RC2: 324: The computation of chi is actually not influenced by smoothing. Other geomorphometric attributes such as stream gradient or ksn, however, are.*

=> This was already noted by reviewer 1. We fully agree with both reviewers, and thank them for this correction!

*RC2: 326: Perhaps also note here the tolerance value applied in the knickpointfinder function.*

=> This is now indicated (line 348).

*RC2: 328: Is there a reference supporting the 3800 m cutoff?*

=> This is justified earlier in the manuscript (with appropriate references), where the general characteristics of the morphology of Bhutan are reported (now lines 236-240).

*RC2: 758: consider to replace "whatever" with "regardless" 765: see comment above*
*RC2: 857: Make sure to format references correctly*
*RC2: 860: remove somehow*
*RC2: 915: "is consistent" rather than "fits"*